

# Phytoplankton growth responses to Asian dust additions in the Northwest Pacific Ocean versus the Yellow Sea

Chao Zhang[1], Huiwang Gao[1*], Xiaohong Yao[1], Zongbo Shi[2], Jinhui Shi[1], Yang Yu[1], Ling Meng[1] and Xinyu Guo[3]

[1]Key Laboratory of Marine Environment and Ecology, Ministry of Education of China, Ocean University of China, Qingdao, China

[2]School of Geography, Earth and Environmental Sciences, University of Birmingham, Birmingham, U.K

[3]Center for Marine Environmental Studies, Ehime University, Japan

*Correspondence to:* hwgao@ouc.edu.cn

**Abstract.** In this study, five on-board microcosm experiments were performed in the subtropical gyre, *Kuroshio* Extension region of the Northwest Pacific Ocean (NWPO) and the Yellow Sea (YS), in order to investigate phytoplankton growth following the addition of artificially modified mineral dust (AM-dust) and various nutrients (nitrogen-N, phosphorus-P, iron-Fe, N+P, and N+P+Fe). The two experiments carried out with AM-dust addition in the subtropical gyre showed a maximum chlorophyll *a* (Chl *a*) concentration increase of 1.7- and 2.8-fold, while the cell abundance of large-sized phytoplankton (>5 µm) showed a 1.8- and 3.9-fold increase, respectively, relative to the controls. However, in the *Kuroshio* Extension region and the YS, the increases in maximum Chl *a* and cell abundance of large-sized phytoplankton following AM-dust addition were at most 1.3-fold and 1.7-fold larger than those in the controls, respectively. A net conversion efficiency index (NCEI) newly proposed in this study, size-fractionated Chl *a*, and the abundance of large-sized phytoplankton were analysed to determine which nutrients contribute to support phytoplankton growth. Our results demonstrate that a combination of nutrients, NP or NPFe as well as other micro-constituents, are responsible for phytoplankton growth in the subtropical gyre following AM-dust addition. Single nutrient addition, i.e., N in the *Kuroshio* Extension region and P/N in the YS, controls the phytoplankton growth following AM-dust addition. In the AM-dust-addition experiments, wherein the increased NP or P were identified to determine phytoplankton growth, the dissolved inorganic P from AM-dust (8.6 nmol L$^{-1}$) was much lower than the theoretically estimated minimum P demand (~20 nmol L$^{-1}$) for phytoplankton growth. These observations suggest that additional supply augments the bioavailable P stock in incubated seawater with AM-dust addition, most likely due to an enhanced solubility of

P from AM-dust or re-mineralization of the dissolved organic P.

## 1 Introduction

Aeolian dust deposition can supply bioavailable nutrients such as nitrogen (N), phosphorus (P), and iron (Fe) to the upper ocean layers (Duce et al., 1991; Jickells et al., 2005; Kanakidou et al., 2012). Observational and modelling studies have

demonstrated that external nutrient input can stimulate the primary productivity, strengthen the nitrogen fixation, alter the phytoplankton size, and potentially enhance the carbon sequestration by the biological pump in the ocean (Mills et al., 2004; Maranon et al., 2010; Liu et al., 2013). A well-recognized aspect of dust deposition is the iron fertilization effect in high-nutrient low-chlorophyll (HNLC) regions (Martin, 1991; Boyd et al., 2007). Recently, many studies have attempted to explore the primary chemicals that promote phytoplankton growth following dust deposition in low-nutrient low-chlorophyll (LNLC)

regions, which cover ~60% of the ocean area worldwide (Moore et al., 2008; Guo et al., 2012; Ridame et al., 2014; Li et al., 2015). For example, dissolved Fe and P from deposited dust were reported to stimulate nitrogen fixation in the oligotrophic region of the eastern tropical North Atlantic Ocean (Mills et al., 2004). In the oligotrophic region of the western North Atlantic Ocean off Barbados, a greater concentration of dissolved N and Fe relative to P, arising from the deposited dust, likely favoured the growth of *Prochlorococcus*, but limited the activity of diazotrophs (Chien et al., 2016). Moreover, the reported positive

effect of dust deposition on the primary production in the central Atlantic Ocean decreased with increasing oligotrophy of the seawater (Maranon et al., 2010). However, studies in areas other than the North Atlantic Ocean and the Mediterranean Sea are scarce.

Dust particles frequently mix with anthropogenic aerosols on their transport pathway to the oceans (Guieu et al., 2010; Shi et

al., 2012; Herut et al., 2016). The response of phytoplankton to the added dust particles mixed with anthropogenic aerosols appeared to be more sensitive in oligotrophic waters than in moderately nutrient-enriched waters (Guo et al., 2012). Unlike in HNLC and LNLC regions of the oceans, the response of phytoplankton to dust deposition in coastal seas that receive a relatively large quantity of nutrients from rivers is poorly understood. A few studies showed that dissolved N from the added

dust likely stimulated phytoplankton growth in the Yellow Sea (YS) (Liu et al., 2013). Added Fe instead of other dissolved

nutrients from atmospheric deposition played an important role in stimulating phytoplankton growth in the Eastern China Sea

(Meng et al., 2016). The complex responses of phytoplankton growth to dust deposition are worth investigation.

The supply of N and P to the ocean surface due to dust deposition are generally higher than the Redfield ratio (N:P = 16)

(Baker et al., 2003; Guo et al., 2012), which reflects the average cellular N:P stoichiometry for oceanic phytoplankton (Arrigo,

2005). Although the deposited P was deficient relative to the demands of the resident phytoplankton in the ocean surface, a

few studies showed that the input of dust could compensate for P deficiency to some extent by stimulating the biogenic

conversion of dissolved organic P (DOP) to dissolved inorganic P (DIP), or a slow release of DIP from dust (Ridame and

Guieu, 2002; Mackey et al., 2012; Krom et al., 2016). This supply of bioavailable P seemingly varies substantially across

different oceanic regions and is affected by many factors, including dust sources and its mixing with anthropogenic pollutants,

P demand, and uptake of nutrients that are co-limiting for phytoplankton in the seawater (Mackey et al., 2012). It remains a

challenge to accurately estimate the supply of bioavailable P for phytoplankton growth, induced by dust deposition in specific

environments.

The arid regions of eastern Asia are the most important mineral dust source to the western Pacific Ocean (Shao et al., 2011).

In spring, strong westerly winds can carry large amounts of mineral dust from Asian continent along a corridor between 25°N

to 45°N, and bring significant amounts of nutrients to downwind areas, including the coastal seas of China and Japan, and the

Northwest Pacific Ocean (NWPO) (Shao et al., 2011). During this long-range transport, the contents of bioavailable nutrients

in the Asian dust might increase through mixing with air pollutants such as sulphur dioxide and nitrogen oxides (Formenti et

al., 2011). The subtropical gyre in the NWPO is a LNLC region with $NO_3^-+NO_2^-$ and $PO_4^{3-}$ concentrations generally maintained

at nanomolar levels (Hashihama et al., 2009). Seawater in the *Kuroshio* Extension region, however, show flexible

characteristics in nutrients, which is ascribed to the confluence of the *Oyashio* current with replete nutrient stocks, and that of

the *Kuroshio* current with impoverished nutrient stocks (Measures et al., 2006; Kitajima et al., 2009). On the other hand, the

YS is a semi-enclosed continental shelf region, which is located in the margin of the NWPO surrounded by the Chinese

mainland and the Korean Peninsula. Seawater in the YS exhibits high nutrient levels but varying nutrient-limiting conditions,

e.g., N or P limitation (Liu et al., 2003). Thus, the NWPO and YS are the ideal zones to explore how Asian dust deposition

influences phytoplankton growth and community size shifts under varying nutrient levels in seawaters.

In this study, we conducted on-board bioassay incubations using artificially modified mineral dust (AM-dust) collected from

the Gobi Desert of China and native phytoplankton assemblages collected from the NWPO and YS, to investigate the effect

of atmospheric nutrients and trace metal inputs on phytoplankton growth. The measured concentrations of Chl $a$ in the nutrient

treatments (i.e., N, P, Fe, N+P, and N+P+Fe additions) against the control were used to determine the nutrient limitation in the

subtropical gyre, the *Kuroshio* Extension region of the NWPO and the YS, respectively. The concentrations of total and size-

fractionated Chl $a$, as well as the abundances and community structures of large-sized phytoplankton ($> 5$ μm) were determined

to study the responses of different-sized phytoplankton to the nutrients supplied by AM-dust. A net conversion efficiency index

(NCEI) is proposed to identify the key nutrient(s) in determining the increase of Chl $a$ concentrations, following AM-dust

addition at different incubation stations. Finally, we analysed the minimum P consumption that supports phytoplankton growth

following AM-dust addition, to estimate the P budget of the added dust.

## 2 Materials and Methods

### 2.1 Preparation of AM-dust

We collected surface soil samples (42.37°N, 112.97°E) from the Gobi Desert, one of the most important sources of dust events

crossing over the YS and NWPO (Ooki and Uematsu, 2005; Shao et al., 2011). The soil samples were crushed, sieved to less

than 20-μm particle size, and freeze-dried (Shi et al., 2009). To account for the aging of dust particles in the atmosphere, we

followed Guieu et al. (2010) by mixing the soil with synthetic cloud water in a cleaned polypropylene bottle, then spread the

solution on a polystyrene tray and evaporated the aqueous phase under clean air flow in a fume hood. A more detailed

modification protocol is presented in the Supplementary Information (Text S1 & Table S1). These treated soil dust particles

are referred to as 'AM-dust'. Clean plastic or plastic-coated materials were used for the preparation of AM-dust to avoid metal

contamination.



## 2.2 Experimental design

Five AM-dust addition bioassay experiments were carried out on-board the *R/V Dongfanghong II* during two cruises in 2014 at stations Ar4, G7, and K4 in the NWPO (Cruise I: March–April), and B7 and H10 in the YS (Cruise II: April–May) (Fig. 1 & Table 1). Based on the baseline Chl *a* concentrations (Table 1), the five stations were redefined as S1 (Ar4), S2 (G7), S3 (K4), S4 (B7), and S5 (H10), in the ascending order. Surface seawater (2–5 m) was collected using acid-washed Teflon-coated Go-Flo bottles mounted on a SeaBird CTD assembly (SBE 9/11, USA) and filtered through a 200-μm acid-washed mesh to

remove larger grazers. Filtered seawater with nine different additions (true replicates, detailed in Table 2) was randomly dispensed into 18 pre-acid-washed and sample-rinsed (three times) Nalgene polycarbonate bottles (20-L each). Water samples in the incubated bottles were collected before the additions, for characterizing the baseline seawater samples, and immediately after the additions, for characterizing the amended seawater (day 0). Surface seawater was pumped into the microcosm equipment, i.e., three large plastic vessels, to stabilize the temperature of the incubation systems (Liu et al., 2013). The

incubation experiments were processed over 9–10 days under natural light. Water samples were collected from incubated bottles at 08:00 am every day except day 1 (i.e., day 2 through day 9/10), for determining Chl *a* and nutrient concentrations, and on certain days for phytoplankton identification and enumeration.

Treatments included AM-dust and single nutrient (N, P, or Fe) additions as well as N+P and N+P+Fe (Table 2). Previous

studies showed that annual deposition flux of Asian dust in the YS and NWPO regions ranged from 10 to 80 g m$^{-2}$ yr$^{-1}$ (Gao et al., 1992; Brown et al., 2005). An extreme dust storm event accompanied by wet precipitation in the YS can lead to a considerable dust deposition flux in this range (Shi et al., 2012). Considering these results, we added 2 mg L$^{-1}$ of the AM-dust to the incubated seawater to simulate deposition due to a strong dust event (20 g m$^{-2}$) in the upper 10 m water layer in the YS (Liu et al., 2013). We added the same amount of AM-dust for incubation experiments in the NWPO for comparison. This

amount of added dust has been widely used in other studies as well (Mills et al., 2004; Maranon et al., 2010). Based on the N-deposition flux in the NWPO (Kim et al., 2014) and an estimate of the N addition by a dust event in the surface water of the

YS (Shi et al., 2012), we added 40 μmol $NaNO_3$ into the 20-L bottles for the N-related treatments (N, N+P, and N+P+Fe

additions), to increase the N concentration by 2 μmol $L^{-1}$. The added P and Fe in incubation systems were 0.2 μmol $L^{-1}$ and 2

nmol $L^{-1}$, respectively, based on the nutrient-addition experiments conducted in the NWPO and YS (Noiri et al., 2005; Liu et

al., 2013) (Table 2).

One-way analysis of variance (ANOVA) was used to assess significant differences in the mean values of the selected

parameters among various treatments, and then Dunnett's test was used to compare these treatments with the control using

SPSS software.


## 2.3 Chl *a* concentration

From each bottle, a 300-mL seawater sample was sequentially filtered through 20-, 2-, and 0.2-μm Whatman polycarbonate

filters to determine size-fractionated Chl *a* concentrations, i.e., pico- (0.2–2 μm), nano- (2–20 μm), and micro-sized (> 20 μm)

Chl *a*. The pigments on the filters were extracted in 90% acetone at -20 °C over 24h in the dark, and measured on a Turner

Designs Trilogy fluorometer (Strickland and Parsons, 1972). The total Chl *a* concentration was calculated as the sum of the

three size-fractionated values.

## 2.4 Inorganic nutrients and trace metals

To determine $NO_3^-$, $NO_2^-$, $NH_4^+$, and $PO_4^{3-}$ concentrations in AM-dust, the 'ultrasonic bath' method described by Liu et al.

(2013) was used to leach the soluble nutrients. A 200-mL seawater sample retrieved from each of the incubation bottles was

filtered through pre-acid-washed cellulose acetate membranes and filtrates, and stored immediately at -20 °C in acid-washed

high-density polyethylene bottles for nutrients analysis in the laboratory. Inorganic nutrients ($NO_3^-$, $NO_2^-$, $NH_4^+$, and $PO_4^{3-}$) in

AM-dust and the incubated seawaters were measured following the automated colorimetric technique described by Grasshoff

et al. (1999) and Ridame et al. (2014), using a QuAAtro Continuous Flow Analyzer (SEAL Analytical). Detection limits of

this instrument were defined as three times the standard deviation of the blank, which corresponds to 30 nM, 7 nM, 80 nM,

and 40 nM for $NO_3^-$, $NO_2^-$, $NH_4^+$, and $PO_4^{3-}$, respectively. Note that the $NH_4^+$ concentrations were determined only for the

leached solution of AM-dust. We followed the method proposed by Hsu et al. (2010) to determine the concentrations of soluble

trace metals leached from AM-dust using an ICP-MS. The recovery yield, accuracy, and detection limit are summarized in

Table S2.


## 2.5 Phytoplankton identification and enumeration

Seawater samples (300-mL volume) were collected from the baseline seawaters at each station and incubated bottles on day 4

from S1 and S2, day 3 from S3 and S5, and day 5 from S4 for the identification and enumeration large-sized phytoplankton (>

5 μm). The sampling dates selected during the incubation experiment were either close to or corresponded exactly with the

days that showed the maximum Chl $a$ concentrations (Figure S1). The sampled seawater was fixed with 1% Lugol's iodine

and stored in dark until microscopic observation in the laboratory (Burson et al., 2016). Before analysis, the preserved samples

were settled for 48 h in the dark and then concentrated to 10-mL volume in glass cylinders. Large-sized phytoplankton was

identified and enumerated using a Nikon ECLIPSE TE2000-U inverted microscope. To illustrate the responses of different

phytoplankton to various additions, the detected phytoplankton species were divided into two major functional groups: diatoms

and dinoflagellates. The dominant species of diatoms found in this study were *Nitzschia* spp., *Chaetoceros* spp., *Thalassiosira*

spp., *Skeletonema* spp., *Cylindrotheca closterium*, and *Rhizosolenia setigera*.

## 2.6 Selection of study period

In this study, Chl $a$ concentrations generally showed a bell-shape growth curve at all stations with the maximum concentration

occurring around 2–5 days (Fig. S1). We focused on analysing the initial 2–5 days, which showed a successive increase in the

incubation period. The duration of 2–5 days for incubation has been widely used in other microcosm experiments (Herut et al.,

2005; Tanaka et al., 2011; Guo et al., 2012; Li et al., 2015) and is supposed to minimize the effects of bottle enclosure and a





possible deviation from the natural environment (Mackey et al., 2012; Coelho et al., 2013).

## 3 Results

### 3.1 Characteristics of baseline surface seawater and AM-dust

Evidently low concentrations of Chl *a* ($\leqslant$ 0.50 µg L$^{-1}$) and nutrients (NO$_3^-$+NO$_2^-$ $\leqslant$ 0.26 µmol L$^{-1}$, PO$_4^{3-}$ = 0.05 µmol L$^{-1}$)

were observed at S1 and S2 in the subtropical gyre of the NWPO (Table 1), indicating oligotrophy (Hashihama et al., 2009).

At S3 in the *Kuroshio* extension and S4 in the YS, Chl *a* and nutrient concentrations were at least 2-fold higher than those at

S1 and S2, indicating an increased trophic level at S3 and S4. For the S5 station in the YS, although the nutrient levels were

comparable to those at S1, Chl *a* concentrations were as high as 2.74 µg L$^{-1}$, close to the conditions of spring bloom in the YS

(Liu et al., 2003; Fu et al., 2009; Liu et al., 2013). The N:P ratios at S1, S2, S3, and S5 were far lower than the Redfield Ratio

(16:1), while that at S4 was as high as 32 (Table 2). Pico-sized Chl *a* accounted for >70% of the total Chl *a* at S1 and S2, but

it decreased to less than ~50% at S3, S4, and S5. Specifically, large-sized phytoplankton abundance showed an increasing

trend from S1 and S2 (< 20 cells mL$^{-1}$) to 79 cells mL$^{-1}$ at S3 and 35 cells mL$^{-1}$ at S4, and to the highest value of 314 cells mL$^{-1}$

at S5. Diatoms dominated the large-sized phytoplankton community at S1, S3, and S5, and a codominance of dinoflagellates

was noted at S2 and S4 (Table 2).

As shown in Table 3, the concentration of dissolved inorganic nitrogen (DIN, i.e. NO$_3^-$+NO$_2^-$+NH$_4^+$) in the AM-dust was 577

µmol g$^{-1}$, which is four times higher than that in the untreated dust. A notable increase in the PO$_4^{3-}$ content (4.3 µmol g$^{-1}$) in

the AM-dust was observed, relative to the untreated dust (1.3 µmol g$^{-1}$). The abundances of DIN, PO$_4^{3-}$, and soluble Fe in the

AM-dust were generally consistent with the values observed in a strong dust event that occurred over the YS in the spring of

2007 (Shi et al., 2012). The N:P ratio in the AM-dust was ~134, far greater than 16 (Redfield Ratio), and similar to those

reported for Asian dust aerosols in previous studies (Shi et al., 2012; Liu et al., 2013; Chien et al., 2016). The NO$_3^-$+NO$_2^-$

concentration in the incubated seawater after AM-dust addition increased up to ~1.13 µmol L$^{-1}$, larger than the baseline NO$_3^-$

+NO$_2^-$ stocks at S1, S2, S3, and S5, but accounted only for approximately 1/3rd of the baseline stock at S4 (Table 2). However,

the increase in PO$_4^{3-}$ (~8.6 nmol/L) following AM-dust additions was negligible, compared with the baseline P stock at each

station (Tables 2 & 3).

### 3.2 Variation in total Chl *a* and nutrient concentrations during incubation experiments

During the successive increase in the incubation period (initial 2–5 days of the incubations, Sect. 2.6), the total Chl *a*

concentration in the controls increased by varying extents at S1–4, while showing a decreasing trend at S5 (Fig. 2). The

maximum Chl *a* concentrations in the controls were 0.60 μg L$^{-1}$ at S1 (day 5), 0.55 μg L$^{-1}$ at S2 (day 2), 1.75 μg L$^{-1}$ at S3 (day

2), 3.24 μg L$^{-1}$ at S4 (day 3), and 2.74 μg L$^{-1}$ at S5 (day 0), respectively. Addition of N+P+Fe induced the largest increases in

Chl *a* among all treatments, and the increases were significant against the controls (Fig. 2, $p < 0.05$) at all the stations. The

maximum Chl *a* concentrations in the N+P+Fe treatments showed a 1.5- to 3.0-fold increase against the controls in the five

incubation experiments. The time-series of Chl *a* in N+P and N+P+Fe treatments almost overlapped at all stations except S2,

where the maximum value of Chl *a* in the N+P treatment was only half of that in the N+P+Fe treatment. Significant increases

in Chl *a* were also observed in N treatments at S1, S2, S3, and S5, and in P treatments at S2 and S4 ($p < 0.05$). The maximum

Chl *a* concentrations showed a 1.2- to 1.6-fold increase in the N treatments at S1, S2, S3, and S5, and 1.4-fold increase in the

P treatments at S2 and S4, relative to the controls. There was no significant increase in the Chl *a* concentration in the rest of

the additions at the five stations. $NO_3^-+NO_2^-$ and $PO_4^{3-}$ concentrations showed a sharp decline in all nutrient treatments, mostly

due to the consumption by the phytoplankton (Fig. 3). The decline in the concentrations of $NO_3^-+NO_2^-$ and $PO_4^{3-}$ relative to

the amended nutrient concentrations on day 0 varied from 43% to 100%.

The addition of AM-dust significantly increased the Chl *a* concentrations ($p < 0.05$) at S1, S2, S3, and S5 (Fig. 2). The

maximum Chl *a* concentrations following AM-dust additions were about 1.7-fold, 2.8-fold, and 1.3-fold of those in the controls

at S1, S2, and S3, respectively. Accordingly, the $NO_3^-+NO_2^-$ and $PO_4^{3-}$ in the AM-dust treatments were consumed completely

at the three stations (Fig. 3). At S4 and S5, the maximum Chl *a* concentration in the AM-dust treatments was comparable to

those in the controls. However, the Chl *a* concentrations remained relatively high in the AM-dust treatments, while showing a

decreasing trend in the controls. The $PO_4^{3-}$ was consumed completely, while there was some residual N in the AM-dust

treatments at S4 and S5 (Fig. 3).





**3.3 Variation in the size-fractionated Chl *a* concentrations during incubation experiments**

When the size-fractionated Chl *a* was examined in various treatments at the five stations, a few notable changes were observed, as highlighted below.

a) In the controls, the dominant contributors to the total Chl *a* were pico-sized cells at S1, S2, S4, and nano-sized cells at S5, while they changed from pico-sized to micro-sized cells at S3 during the successive increase in the incubation period (Fig. 4). Addition of AM-dust increased the Chl *a* concentrations of all sizes and the dominant contributor was consistent with those in the controls at the five stations. However, the magnitude of Chl *a* increase was the highest for micro- or nano-sized cells following AM-dust additions. For example, the largest increase in maximum Chl *a* occurred in micro-sized cells at S1 (2.0-fold), S2 (4.4-fold), S4 (1.4-fold), and S5 (1.6-fold), and in nano-sized cells at S3 (1-3 fold), compared with the controls.

b) Size-fractionated Chl *a* in N+P and N+P+Fe additions showed the largest and similar increases at S1, S3, S4, and S5, especially in the micro-sized range, where the maximum Chl *a* concentrations showed a 1.2- to 6.5-fold increase against the controls (Fig. 4). For station S2, the micro- and nano- sized Chl *a* showed higher increases in N+P+Fe treatments than in N+P treatments, whereas the inverse was true for the pico-sized Chl *a*. Addition of N or P alone also led to an increase in size-fractionated Chl *a* to some extents at all stations. The size-fractionated Chl *a* in N treatments at S1, S3, and S5 increased noticeably, e.g., the maximum Chl *a* showed the largest increase in nano-sized cells by a factor of 1.6 at S1, and 1.4 at S3, and in micro-sized cells by a factor of 2.2 at S5, relative to the controls. A clear increase was also observed in the P treatment at S2 and S4, e.g., the maximum Chl *a* concentrations showed a 1.8-fold (Nano-) and 1.5-fold (Micro-) increase against the controls (Fig. 4). There was no noticeable increase in any of the size-fractionated Chl *a* following Fe addition at S2 and S4.

c) The results of size-fractionated Chl *a* demonstrated that the micro- and nano-sized phytoplankton generally showed a stronger response than pico-sized cells following AM-dust and various nutrient additions, although the extent of increase varied among the stations.



**3.4 Variation in large-sized phytoplankton abundance and community during incubation experiments**

After 2–5 incubation days, diatoms generally dominated the large-sized phytoplankton community in all treatments at the five

stations. *Chaetoceros* spp. and *Nitzschia* sp. comprised the largest fraction of diatoms in all treatments at S1, S2, and S3. For

stations S4 and S5, the dominant species of diatoms did not change compared with those in the baseline seawaters (Fig. 5).

Specifically, the cell abundance of large-sized phytoplankton in the controls increased to 37 cells mL$^{-1}$ at S1, 39 cells mL$^{-1}$ at

S2, 422 cells mL$^{-1}$ at S3, 93 cells mL$^{-1}$ at S4, and 643 cells mL$^{-1}$ at S5. Similar to the response of Chl *a*, additions of N+P and

N+P+Fe increased the phytoplankton abundances noticeably in all treatments by about 2.5- to 2.7-fold at S1 and S5, 3.1- to

3.3-fold at S3, and 2.7-fold at S4, relative to the controls. For single-nutrient treatments, N additions at S1, S3, and S5, and P

additions at S4 induced the highest increases in cell abundance, with 1.5-, 1.6-, 2.3-, and 1.9-fold higher values, respectively,

than those in the controls. The cell abundance of large-sized phytoplankton in AM-dust treatments relative to the controls

increased by 1.8-fold at S1 and 1.7-fold at S3 and S5, but showed a negligible increase at S4. Moreover, the increases following

AM-dust additions were comparable, and sometimes even larger, relative to those in the N treatments at S1, S3, and S4, but

lower than the latter at S5. At S2, the cell abundance with AM-dust addition increased by 3.9-fold against the controls, which

was comparable to those in the N+P treatments (Fig. 5).

**4 Discussion**

**4.1 Nutrient limitation in the NWPO and YS**

Building on the results mentioned above (Fig. 2 and Table 2), we summarized the nutrient limiting status at the five stations.

At S1, phytoplankton growth was very likely co-limited by N and P, because: 1) N addition induced significant increase in Chl

*a* against the controls ($p < 0.05$), 2) there were significant increases in Chl *a* following N+P additions, compared with the N

treatments ($p < 0.05$), and 3) there were no significant differences in Chl *a* between N+P and N+P+Fe treatments ($p > 0.05$).

At S2, a significant increase in Chl *a* was observed in the P, N+P, and N+P+Fe treatments ($p < 0.05$) against the controls, and

addition of N+P and N+P+Fe induced a significantly larger response, relative to the P treatments ($p < 0.05$). Besides, Chl *a*

concentrations in the N+P+Fe treatments were significantly larger than in the N+P treatments on days 4–5. Therefore, we

concluded that phytoplankton at S2 were primarily co-limited by N, P, and Fe (Fig. 2). At S3, N addition induced a significant

increase in Chl *a* compared with the controls, while an even larger significant increase was observed upon N+P (+Fe) addition,

relative to the N treatments. Thus, phytoplankton was primarily co-limited by N and P at S3. Accordingly, $NO_3^-+NO_2^-$ and

$PO_4^{3-}$ concentrations in N+P (+Fe) treatments at the three stations decreased by 73%–100%, larger than the decline in $NO_3^-$

$+NO_2^-$ concentrations after adding N alone, and $PO_4^{3-}$ concentrations after adding P alone (43%–70%) (Fig. 3). Such co-limiting

conditions have been widely reported in the oceans, especially in oligotrophic regions such as Eastern Mediterranean Sea

(Tanaka et al., 2011) and South China Sea (Guo et al., 2012). Moore et al. (2013) argued that low abundances of N and P in

oligotrophic environments are likely to simultaneously reach the limiting levels for phytoplankton growth. Note that

phytoplankton at S3 in the *Kuroshio* Extension showed a greater response to N rather than P addition. The intrusion of

*Kuroshio–Oyashio* transition water from the north and *Kuroshio* water from the southwest, both carrying a certain amount of

macronutrients, typically with N:P < 16, likely changed the nutrient stocks of the seawaters at S3 (Whitney, 2011; Guo et al.,

2012; Yatsu et al., 2013). A low primary productivity in the *Kuroshio* Extension region during the spring time was also

indicated by the low availability of nitrate in the seawater (Nishibe at al., 2015).


Similarly, we also found that P and N were the primary limiting nutrients at S4 and S5, respectively (Fig. 2). This is consistent

with the baseline N:P ratios at S4 (>16) and S5 (<16) (Table 2). In fact, the $PO_4^{3-}$ after adding P alone at S4, and $NO_3^-+NO_2^-$

after adding N alone at S5, decreased by 80% and 100%, respectively, during the successive increase in incubation period (Fig.

3). Complex hydrographic conditions create a large spatiotemporal variation in nutrient concentrations in the YS (Liu et al.,

2003). The riverine input and atmospheric deposition with a high N:P (>16) lead to relatively P-deficient conditions, while the

rapid uptake by phytoplankton characterized by a lower N:P (<16) during the bloom period likely accelerates the decline of P

concentrations in the surface seawaters (Liu et al., 2003; Arrigo, 2005; Liu et al., 2013). In general, the N:P ratios during the

spring time in the coastal waters (near S4) are higher than 16 and lower than 16 in the central waters (near S5) of the YS (Fu

et al., 2009). The primary P- or N-limiting conditions in the surface seawaters of the YS have also been widely reported in

previous studies (Wang et al., 2003; Liu et al., 2013).



## 4.2 Positive effects of AM-dust on phytoplankton growth in the NWPO and YS

It was clear from the data that AM-dust addition increased the Chl $a$ concentrations and large-sized phytoplankton abundance to varying extents at the five stations (Fig. 2, 4&5). To determine which nutrients in the AM-dust treatments supported phytoplankton growth, we introduced a net conversion efficiency index (NCEI) to illustrate the differences in N utilization efficiency between the AM-dust treatments and N, N+P, and N+P+Fe treatments. It was calculated using the following equation:

$$NCEI = \frac{\sum_{i=0}^{t}(Chl\ a_{Ti} - Chl\ a_{Ci})}{\Delta N} \tag{1}$$

where Chl $a_{Ti}$ and Chl $a_{Ci}$ represent the Chl $a$ concentrations on the $i$th day (i.e., day 0, day 2–day 3/5) in the treatments and the control ($\mu g\ L^{-1}$), respectively. $\Delta N$ is the decreased N concentration in the treatment minus that in the control during the successive increase in the incubation period ($\mu mol\ L^{-1}$). The decreased N concentration is calculated as the difference in $NO_3^-$ $+NO_2^-$ before and after the initial increase period. A large NCEI value represents a positive effect of the added N on phytoplankton growth, and a value close to zero represents no effect. Larger the NCEI value, higher is the N utilization efficiency by the phytoplankton.

Note that $NO_3^-$+$NO_2^-$ was added in the AM-dust treatments at a concentration of 1.13 $\mu mol\ L^{-1}$ (Table 3). However, the increase in $NO_3^-$+$NO_2^-$ concentration was only 0.59–0.85 $\mu mol\ L^{-1}$, immediately after the addition and mixing steps (i.e. on day 0) at the five stations, which was about 52%–75% of the added $NO_3^-$+$NO_2^-$. Similar trends were also observed for $NO_3^-$+$NO_2^-$ and $PO_4^{3-}$ concentrations in the nutrient treatments (Fig. 3). The missing N and P were likely adsorbed on the bottle walls and/or suspended particles in the solution (Liu et al., 2013), but these 'missing parts' had the potential to be released into the solution again, when the nutrients were consumed by the phytoplankton. Thus, the added N and P, plus the baseline concentrations in the seawaters, were considered the amended concentrations on day 0, which were used for calculating the NCEI.

The added N in AM-dust and N-related treatments generally stimulated phytoplankton growth to varying extents, as indicated by the large NCEI values (Fig. 6). Note that the NCEI in N treatments (hereafter NCEI$_N$) at S4 was around 0, because the phytoplankton was primarily limited by P. Besides, the significant difference in NCEI among various nutrient treatments can

also reflect the roles played by other nutrients such as P and Fe in affecting the N utilization efficiency. The NCEI in the N+P+Fe treatments were $1.6 \pm 0.1$ at S1, $1.5 \pm 0.1$ at S3, $2.7 \pm 0.1$ at S4, and $1.5 \pm 0.2$ at S5, comparable with those in N+P treatments (Fig. 6). This suggested that the effect of added Fe on the N-utilization efficiency was negligible at the four stations.

However, for S2, although the added Fe alone had no obvious influence on the Chl $a$ concentration (Fig. 2), the significantly

higher value of $NCEI_{N+P+Fe}$ ($1.8 \pm 0.0$) than the $NCEI_{N+P}$ ($1.6 \pm 0.1$) ($P < 0.05$) demonstrated that Fe contributed to increase the N-conversion efficiency during Chl $a$ generation (Twining & Baines, 2015). In addition, the values of $NCEI_{N+P}$ and $NCEI_{N+P+Fe}$ were significantly larger than those of $NCEI_N$ at S1, S2, S3, and S4 (Fig. 6), indicative of the fact that added P contributed to the increased N utilization efficiency at these stations. Indeed, P is an important part of the ribosome and can promote cell division (Arrigo, 2005), which likely increased the NCEI values to some extent. Liu et al. (2013) also reported

the excessive P relative to N (N:P<16) in incubated seawater could increase the biomass. There were no significant differences in the NCEI values between the various treatments at S5, demonstrating that the effects of other nutrients (i.e., P and Fe) on N-conversion efficiency were negligible compared with N.

The NCEI in the AM-dust treatments was $1.3 \pm 0.1$ and $1.3 \pm 0.0$ at S1 and S4, respectively. These two values were significantly

higher than those in the N treatments (Fig. 6). Considering the P-limiting conditions at both stations, we concluded that the external supply of P associated with the AM-dust input likely also played a role in increasing the NCEI. Moreover, for station S2, the $NCEI_{AM-dust}$ of $2.4 \pm 0.1$ was significantly higher than $NCEI_{N+P+Fe}$ ($1.8 \pm 0.0$), $NCEI_{N+P}$ ($1.6 \pm 0.1$), and $NCEI_N$ ($0.3 \pm 0.1$). Micro-nutrients such as Zn and Mn (Coale, 1991; Jakuba et al., 2008; Saito et al., 2008; Sunda, 2012) from the AM-dust may have contributed to the phytoplankton growth in the incubations at S2. This potential synergistic effect is worthy of further

investigation. For stations S3 and S5, there was no significant difference in the NCEI between the AM-dust (i.e., $1.2 \pm 0.2$ at S3, $1.1 \pm 0.4$ at S5) and N treatments (i.e., $1.1 \pm 0.0$ at S3, $1.2 \pm 0.2$ at S5) (Fig. 6), indicating a negligible effect of the dissolved P from AM-dust on NCEI.

Fig. 7 shows the changes in the three size-fractionated Chl $a$ concentrations in the treatments, relative to the corresponding

controls. Generally, the relative changes in the size-fractionated Chl $a$ in AM-dust treatments showed a decreasing trend with

decreasing cell size, i.e., micro- ≥ nano- ≥ pico-sized cells, consistent with the pattern observed in N+P+Fe treatments. Previous incubation experiments have also showed that addition of Asian aerosols could shift the phytoplankton size towards larger cells (Guo et al., 2012; Liu et al., 2013). Indeed, micro- and nano-sized cells, as indicated by faster uptake rates for nutrients and higher biomass-specific production rates, have a growth advantage relative to the pico-sized cells, when the

added materials relieve the nutrient-limiting pressures (Cermeno et al., 2005; Maranon et al., 2007&2012). Besides, the relative changes in micro- and nano-sized Chl *a* in AM-dust treatments were generally higher than in the N treatments at S1, S2, and S4. Especially for station S2, the relative changes in micro- and nano-sized Chl *a* in the AM-dust treatments were even larger than in the N+P treatments, and comparable to those in the N+P+Fe treatments (Fig. 7). These results demonstrated the importance of increased bioavailability of P at S1 and S4, and that of P and Fe at S2, following AM-dust addition, which

supported the growth of micro- and nano-sized phytoplankton.

When the cell abundances of large-sized phytoplankton were considered, we also found that the addition of AM-dust induced larger increases than N additions after certain incubation days at S1 and S2, as characterized by co-limitation of multiple nutrients. Especially at S2, the increase in large phytoplankton abundance was even larger than in the N+P treatments and

lower than in the N+P+Fe treatments. However, for station S3, characterized by NP co-limitation, the changes in the cell abundance of large-sized phytoplankton in AM-dust treatments were comparable to those in the N treatments. These results indicated that the supply of NP and NPFe induced by AM-dust addition, likely contributed to the increase in large-sized cell abundance at S1 and S2, respectively, whereas the supplied N from AM-dust likely determined the growth of large-sized phytoplankton at S3, because the supply of P from the AM-dust was negligible relative to the baseline P stocks (~0.11 μmol

L$^{-1}$). For station S5, characterized by N limitation alone, the addition of AM-dust provided the limiting nutrient to increase the cell abundance of large-sized phytoplankton (Fig. 5).

A notable change in the diatom cell abundance in response to the addition of AM-dust was observed at S1, S2, S3, and S5 (Fig. 5). Indeed, when the nutrient-limitation pressures were relieved, diatoms tended to prosper rapidly, which was ascribed to their

fast uptake rates and quick metabolic responses to nutrient additions (Fawcett and Ward, et al., 2011; Franz et al., 2012). The



dominant species of diatoms in our study were similar to the results from other incubation or observation experiments in the North Pacific and YS (Noiri et al., 2005; Aizawa et al., 2005; Sun et al., 2013). For station S4, we observed higher contributions of dinoflagellates (40%–48%) to the large-sized phytoplankton community, in terms of cell abundance for AM-dust and N additions, compared with the P, N+P, and N+P+Fe treatments (18%–28%) (Fig. 5). This was closely related to the fast growth of diatoms as a result of the relief from P-limiting conditions after the addition of P, N+P, and N+P+Fe. Besides, it has been reported that dinoflagellates with lower demands and higher P absorption efficiency have an advantage of surviving in P-deficient conditions, compared with diatoms (Egge, 1998; Zhou et al., 2008). Hence, in contrast to P-related treatments, the supply of bioavailable P from AM-dust was not enough to support the growth of diatoms at S4 (Fig. 5).

### 4.3 Bioavailable P in the AM-dust addition experiment

The increase in bioavailable P concentration following AM-dust addition played an important role in stimulating phytoplankton growth at S1, S2, and S4, as mentioned above (Sect. 4.2). However, we did not detect a marked increase in the $PO_4^{3-}$ concentrations in incubated seawaters, following AM-dust additions, during the successive increase in incubation period at the three stations (Fig. 3). This could be ascribed to the gradual bioavailable P released over time and its rapid uptake by phytoplankton with a high P demand (Mackey et al., 2012), and the easy adsorption of $PO_4^{3-}$ by particles or bottle walls (Liu et al., 2013), which likely increases of $PO_4^{3-}$ concentration in the incubated seawaters below the detection limit. In order to quantify the increased bioavailable P concentrations following AM-dust addition, we analysed the correlation of the consumed ratios of N:P (hereafter, $C_{N:P}$, calculated as the ratios of the difference in $NO_3^-+NO_2^-$ and $PO_4^{3-}$ concentrations at the beginning and the end of the successive increase in incubation period), based on the supply ratios of N:P (hereafter, $S_{N:P}$, i.e., the amended ratios of $NO_3^-+NO_2^-$ and $PO_4^{3-}$ concentrations in the incubated seawaters on day 0) in AM-dust and N treatments.

The relationships between $C_{N:P}$ and $S_{N:P}$ in the control and N, P, and N+P (+Fe) treatments at all stations are plotted in Fig. 8. We found that $C_{N:P}$ significantly increased with $S_{N:P}$, and the same trend was observed at each station (Fig. S2). Klausmeier et al. (2004, 2008) pointed out that the phytoplankton in nutrient-rich environments tends to exhibit an exponential/bloomer





growth with a requirement of N:P ratio in a relatively constant value, regardless of the varying nutrient supply ratios. However,

when the phytoplankton tend to sustain a competitive equilibrium status in nutrient-limited environments, e.g., N or P, their

requirements of N:P ratios generally increase with the nutrient supply ratios. In our study, the phytoplankton at the five stations

were limited by different nutrients, and thus, tended to sustain a competitive equilibrium status. As the provided DIN following

the AM-dust addition was 1.15 μmol N L$^{-1}$, lower than the amount of N when added alone (2 μmol N L$^{-1}$), the $S_{N:P}$ in AM-dust

treatment was lower than that in N treatments. Thus, $C_{N:P}$ in both treatments at each station would follow the pattern: $C_{N:P}$ in

AM-dust treatments ⩽ $C_{N:P}$ in N treatments. The consumed N in AM-dust treatments, as well as N and P in N treatments,

can be determined on basis of nutrient measurements at the beginning and the end of the successive increase in the incubation

period (Fig. 3).

If we consider the $C_{N:P}$ in AM-dust treatments is equal to that in N treatments, the minimum consumed P in the AM-dust

treatments was estimated by Eq. (2), yielding values of 76 nmol L$^{-1}$ at S1, 46 nmol L$^{-1}$ at S2, 98 nmol L$^{-1}$ at S3, 122 nmol L$^{-1}$

at S4, and 20 nmol L$^{-1}$ at S5. Deducting the P concentrations in the baseline seawaters (Table 2), the minimum bioavailable P

supplied by AM-dust additions were estimated to be 22 nmol L$^{-1}$ at S1, -5 nmol L$^{-1}$ at S2, -6 nmol L$^{-1}$ at S3, 18 nmol L$^{-1}$ at S4,

and -20 nmol L$^{-1}$ at S5. Note that the dissolved P from AM-dust was as low as 8.6 nmol L$^{-1}$ in the incubation bottles, which

could not meet demands of phytoplankton at S1 and S4. The negative values of the minimum bioavailable P induced by AM-

dust additions at S2, S3, and S5 demonstrated that the P concentrations in baseline seawaters likely supported the growth of

phytoplankton, which might be related to other limiting nutrients (except N and P) dissolved from the AM-dust at S2, limited

supply of P from AM-dust at S3 (Sect. 4.2), and primary N-limiting conditions at S5 (Sect. 4.1).

$$CP_{AD} = \frac{CN_{AD} \times CP_N}{CN_N} \tag{2}$$

where CP and CN are the consumed bioavailable P and NO$_3^-$+NO$_2^-$ concentrations (μmol L$^{-1}$) in the incubation systems during

the successive increase in incubation period, respectively. Subscript means AM-dust ('AD') or N treatments.

As described above, indirect supply, other than the dissolved bioavailable P (8.6 nmol L$^{-1}$), induced by the AM-dust, might

exist. The dissolved organic phosphorus (DOP) is considered a significant portion of the dissolved P pool in oceans, especially

in surface seawater (Paytan and McLaughlin, 2007). Dust input has been reported to induce the biological mineralization of

DOP to DIP under P-deficient conditions, and consequently, increased P bioavailability (Mackey et al., 2012). Trace elements

such as Zn and Co are important components of the metalloenzyme alkaline phosphatase, which could be provided by the dust

input and facilitates the acquisition of bioavailable P from the DOP pool (Cembella et al., 1982; Jakuba et al., 2008). DIP in

the seawater would even accumulate if the demand of phytoplankton for P was smaller than the total amount of DIP released

from the dust as well as that from the mineralized from DOP (Mackey et al., 2012). Note that the gradual enhanced solubility

of atmospheric P with the duration of exposure in seawater was also reported to contribute additional bioavailable P for

phytoplankton growth (Ridame and Guieu, 2002; Mackey et al., 2012). The determination of DOP and total P concentrations

in the incubated seawaters and P contents in the phytoplankton cells would be helpful for examining the sources of P in future

studies.

**5 Conclusion**

In this study, phytoplankton growth was found to be limited by two or more nutrients (i.e., NP or NPFe) in the NWPO, and by

a single nutrient (i.e., N or P) in the YS. In the subtropical gyre of the NWPO, the addition of AM-dust provided NP or NPFe

and micro-constituents to stimulate phytoplankton growth. In comparison with the controls, the maximum Chl *a* following

AM-dust addition showed a 1.7- and 2.8-fold increase, while the cell abundance of large-sized phytoplankton showed a 1.8-

and 3.9-fold increase. As the increased P from AM-dust was negligible in comparison with the baseline P stocks, the dissolved

N from AM-dust, thus, primarily supported the phytoplankton growth in the *Kuroshio* extension. The maximum Chl *a*

concentrations and cell abundance of large-sized phytoplankton following AM-dust addition were 1.3-fold and 1.7-fold larger,

respectively, than those in the controls. In the YS, the increased P or N by AM-dust additions primarily contributed to the

growth of phytoplankton. The Chl *a* concentrations in AM-dust treatments were generally higher than those in the controls,

although the differences in maximum Chl *a* were negligible in both groups. The increase in the cell abundance of large-sized

phytoplankton was < 1.7-fold, compared with the controls. Comparing the difference in consumed N:P between the AM-dust

and N treatments, we found that the directly supplied bioavailable P by AM-dust in the incubated seawaters was not enough

to support phytoplankton growth in the YS, which is characterized by P limitation, and in the subtropical gyre, which is characterized by NP co-limitation. This suggests that there are missing sources of P in the incubations, which may be explained

by the enhanced solubility of P from AM-dust and/or mineralization of DOP in the seawaters. Besides, the addition of AM-dust had a potential to shift the phytoplankton towards larger cells at all incubation stations, although it did not change the dominant taxa of phytoplankton assemblages. In general, larger positive responses of phytoplankton induced by combined nutrients than by single nutrient from the AM-dust were observed in our study. This is likely related to the varying nutrient levels, community structures of phytoplankton in the baseline seawaters, and the input of nutrients following AM-dust addition.


Our study proposes the importance of increase in bioavailable P stock for phytoplankton growth following AM-dust addition. This would help us better understand the effects of dust deposition on P cycles in the ocean. In view of the increasing anthropogenic N deposition in the NWPO and YS, due to continuously strong $NO_x$ emissions in the eastern Asia (Kim et al., 2011; Kim et al., 2014), the increase in bioavailable P stock induced by dust deposition might be even more important in

phytoplankton growth in the future. Moreover, further investigations are needed to better understand the differential effects of increase in bioavailable P stock as a result of the atmospheric deposition on phytoplankton growth in the coastal seas and open oceans. It is noted that the stimulation of large-sized phytoplankton growth due to the input of AM-dust might enhance the carbon storage in the deep ocean, as the sinking rate of large-sized cells in the water column are higher than the pico-sized ones (Bach et al., 2012).


**Acknowledgments.** This work was funded by National Natural Science Foundation of China (NSFC) (41210008: Gao), Major State Basic Research Development Program of China (973 Program) (2014CB953701: Gao), and NSFC and Royal Society travel grant (4141101141: Gao and Shi).

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




**Table 1.** Experimental information and baseline conditions at the sampling stations

|  | Ar4 (S1) | (G7) S2 | K4 (S3) | B7 (S4) | H10 (S5) |
|---|---|---|---|---|---|
| **Sampling date** | 2014.3.23 | 2014.4.5 | 2014.4.14 | 2014.5.9 | 2014.4.29 |
| **Incubation time** | 10 days | 9 days | 9 days | 10 days | 9 days |
| **Sampling location** | 29.5°N, 142.5°E | 30.0°N, 148.5°E | 34.0°N, 144.0°E | 37.00°N, 123.17°E | 35.0°N, 124.0°E |
| **Temperature(°C)** | 18.9 | 18.3 | 18.6 | 9.84 | 18.9 |
| **Mixed layer depth (MLD)** | 150 | 82 | 103 | - | - |
| **Chl $a$ (µg L$^{-1}$)** | 0.24 | 0.50 | 1.28 | 1.58 | 2.74 |
| **NO$_3^-$+NO$_2^-$ (µmol L$^{-1}$)** | 0.26 | 0.04 | 0.51 | 3.20 | 0.28 |
| **PO$_4^{3-}$ (µmol L$^{-1}$)** | 0.05 | 0.05 | 0.11 | 0.10 | 0.04 |
| **N:P (µmol:µmol)** | 5 | <1 | 5 | 32 | 7 |
| **Micro-sized Chl a (%)** | 6 | 8 | 28 | 14 | 27 |
| **Nano-sized Chl a (%)** | 18 | 19 | 23 | 36 | 37 |
| **Pico-sized Chl a (%)** | 76 | 73 | 49 | 50 | 36 |
| **Large-sized phytoplankton abundance (cell mL$^{-1}$)** | 17 | 10 | 79 | 35 | 314 |
| **Diatoms (cell mL$^{-1}$)** | 12 | 6 | 74 | 16 | 286 |
| **Dinoflagellates (cell mL$^{-1}$)** | 5 | 4 | 5 | 19 | 28 |



**Table 2.** Treatments to the bioassay incubation experiments

| Incubation No. | Treatments | Amended concentrations |
|:---:|:---:|:---:|
| 1 | Control | - |
| 2 | AM-dust | 2 mg L$^{-1}$ |
| 3 | NaNO$_3$ | 2 μmol L$^{-1}$ |
| 4 | NaH$_2$PO$_4$ | 0.2 μmol L$^{-1}$ |
| 5* | FeCl$_3$ | 2 nmol L$^{-1}$ |
| 6 | NaNO$_3$+NaH$_2$PO$_4$ | 2 μmol L$^{-1}$+0.2 μmol L$^{-1}$ |
| 7 | NaNO$_3$+NaH$_2$PO$_4$+FeCl$_3$ | 2 μmol L$^{-1}$+0.2 μmol L$^{-1}$+2 nmol L$^{-1}$ |

*Only at stations S2 and S4.





**Table 3.** Nutrient and soluble trace-metal contents in artificially modified and untreated dust, and increased corresponding concentrations in the incubated seawaters amended with AM-dust.

| | Nutrients (µmol g⁻¹) | | | | Soluble trace metals (µg g⁻¹) | | | | | | |
|---|---|---|---|---|---|---|---|---|---|---|---|
| | $NO_3^-$ | $NO_2^-$ | $NH_4^+$ | $PO_4^{3-}$ | Fe | Mn | Cu | Cd | Pb | Co | Zn |
| **AM-dust** | 532.9 | 34.2 | 10.3 | 4.3 | 473.1 | 413.5 | 0.23 | 0.04 | 0.24 | 2.58 | 4.27 |
| **Untreated dust** | 133.2 | 8.6 | 10.5 | 1.3 | 22.7 | 1.66 | ND[a] | ND | ND | ND | 0.03 |
| **Increased concentrations amended with AM-dust (nmol L⁻¹)** | 1066 | 68.4 | 20.6 | 8.6 | 16.90 | 15.05 | 7.2E-3 | 7.1E-4 | 2.3E-3 | 0.09 | 0.13 |

[a]'ND' means no detection.

    [b]Calculated by dividing the added amounts of nutrient and soluble trace metals with AM-dust input by the incubation volume of 20 L.



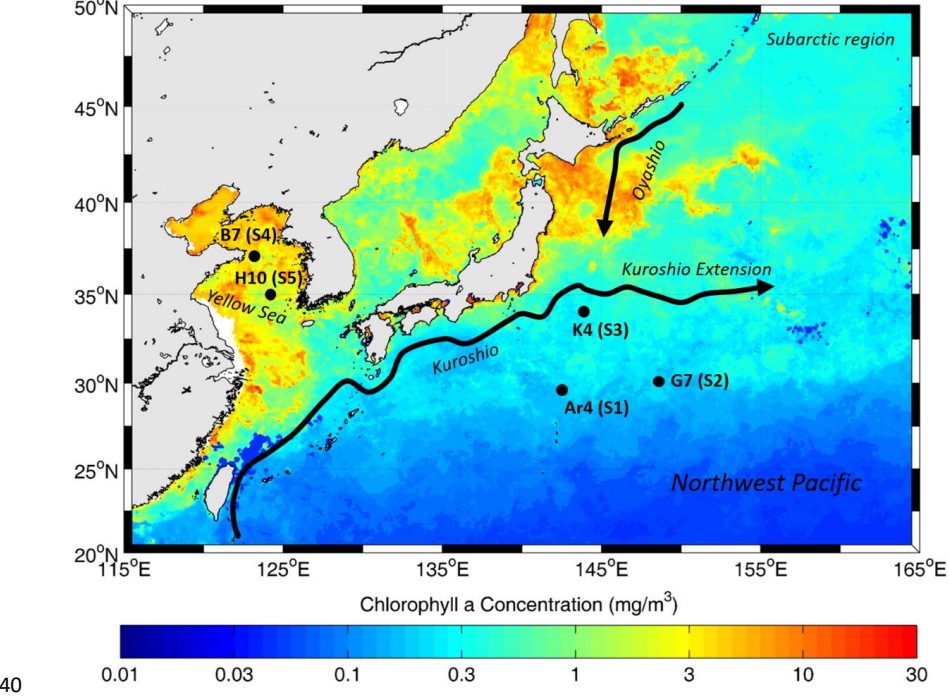


**Figure 1.** Locations of the five seawater-collection stations for the microcosm experiments (The base map reflects an average composite of Chl *a* concentrations in March-May, 2014 obtained from the NASA website (https://modis.gsfc.nasa.gov/data/dataprod/chlor_a.php)




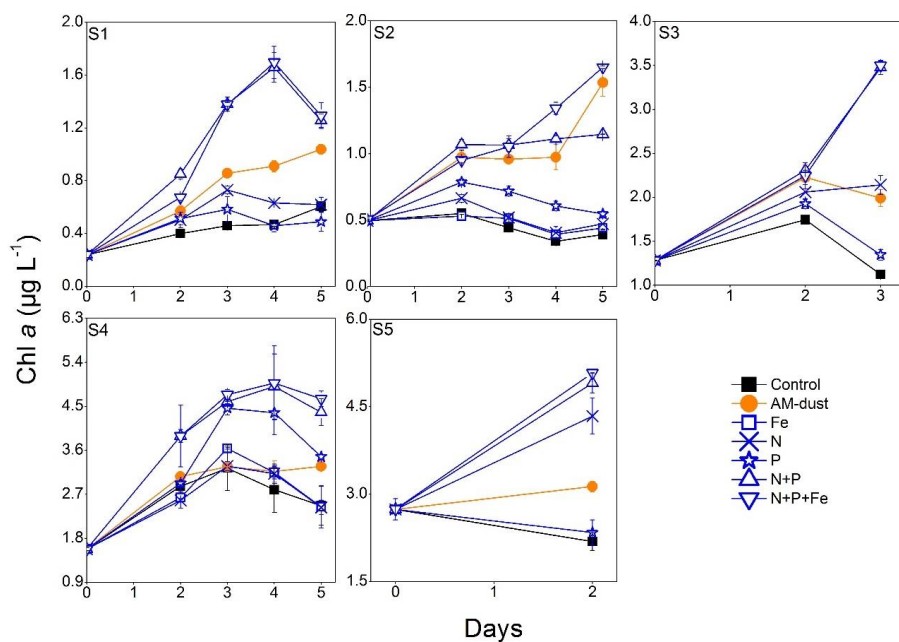

**Figure 2.** Changes in Chl *a* during the successive increase in the incubation period at each station




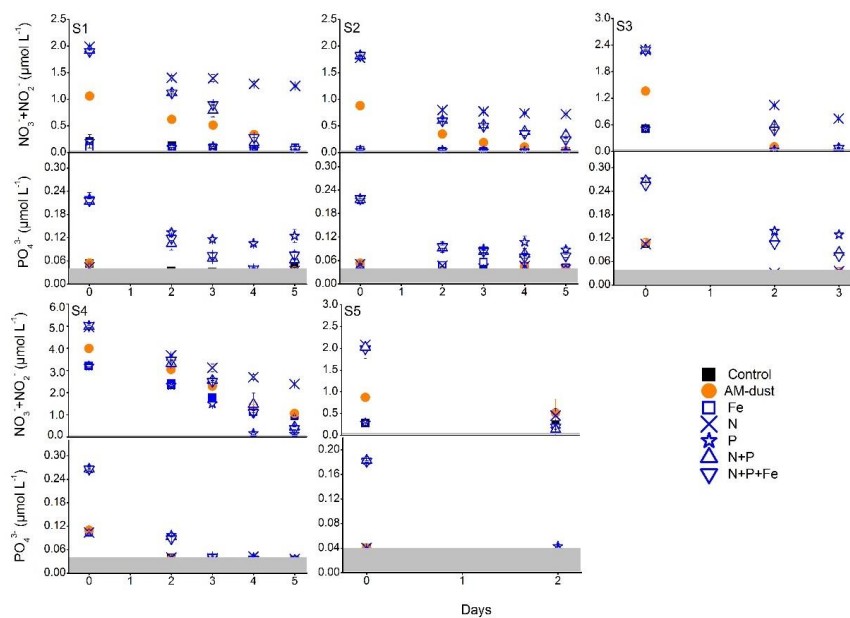

**Figure 3.** Changes in $NO_3^-$+$NO_2^-$ and $PO_4^{3-}$ concentrations during the successive increase in the incubation period at each station. Shaded areas mean the values below the detection limit



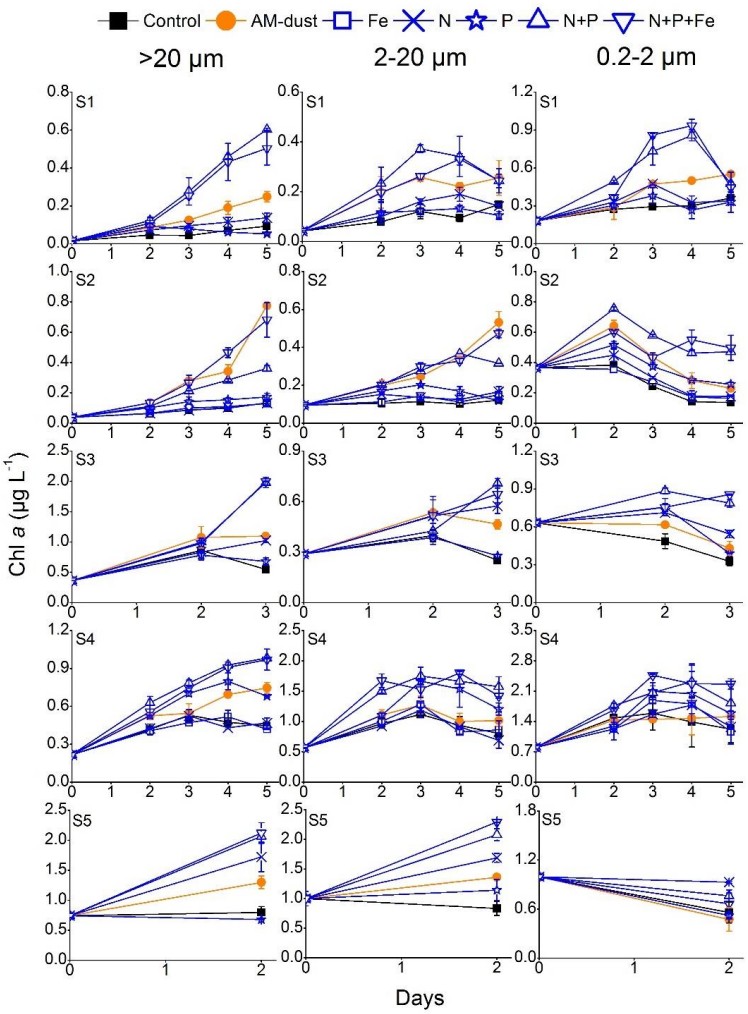

**Figure 4.** Changes in size-fractionated Chl *a* during the successive increase in the incubation period at each station





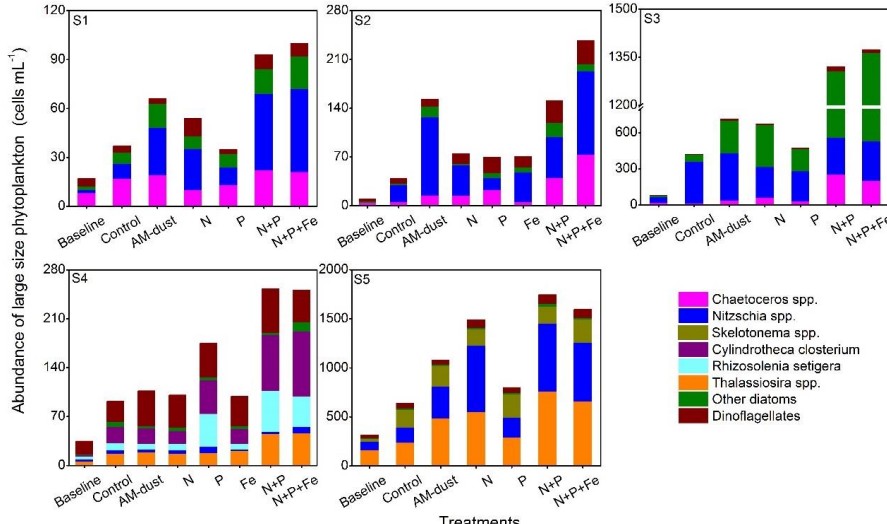

**Figure 5.** Changes in the taxonomic structure of large-sized phytoplankton community before (i.e., 'baseline') and
during (i.e., 'control' and various treatments) the successive increase in the incubation period at each station



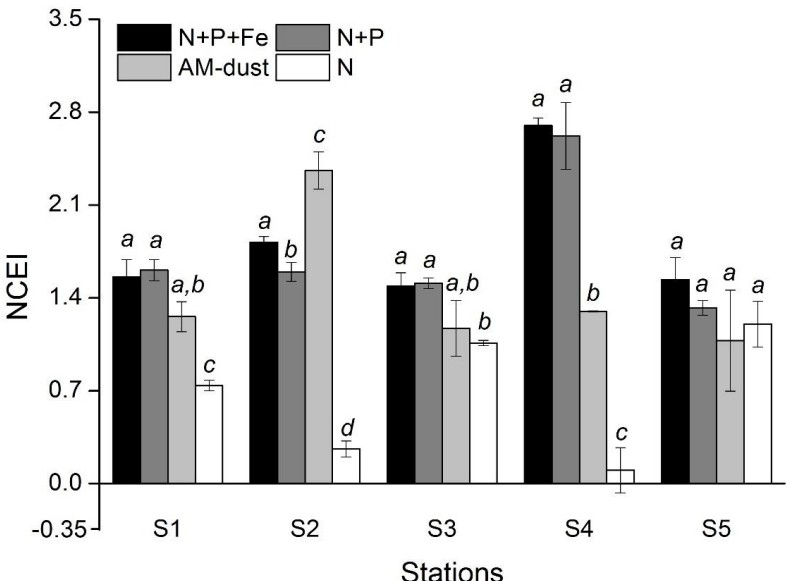

**Figure 6.** The net conversion efficiency index (NCEI) of N conversion to Chl *a* in AM-dust and various nutrient additions during the successive increase in the incubation period at each station. Means of NCEI in various treatments that are significantly different are labelled with a different letter ($p < 0.05$). For example, the NCEI in two treatments labelled with the same letter ('a' or 'a, b' and 'a/b') indicates that there is no significant difference between them, whereas different letters, ('a' and 'b' or 'a, b' and 'c') indicate a significant difference between them



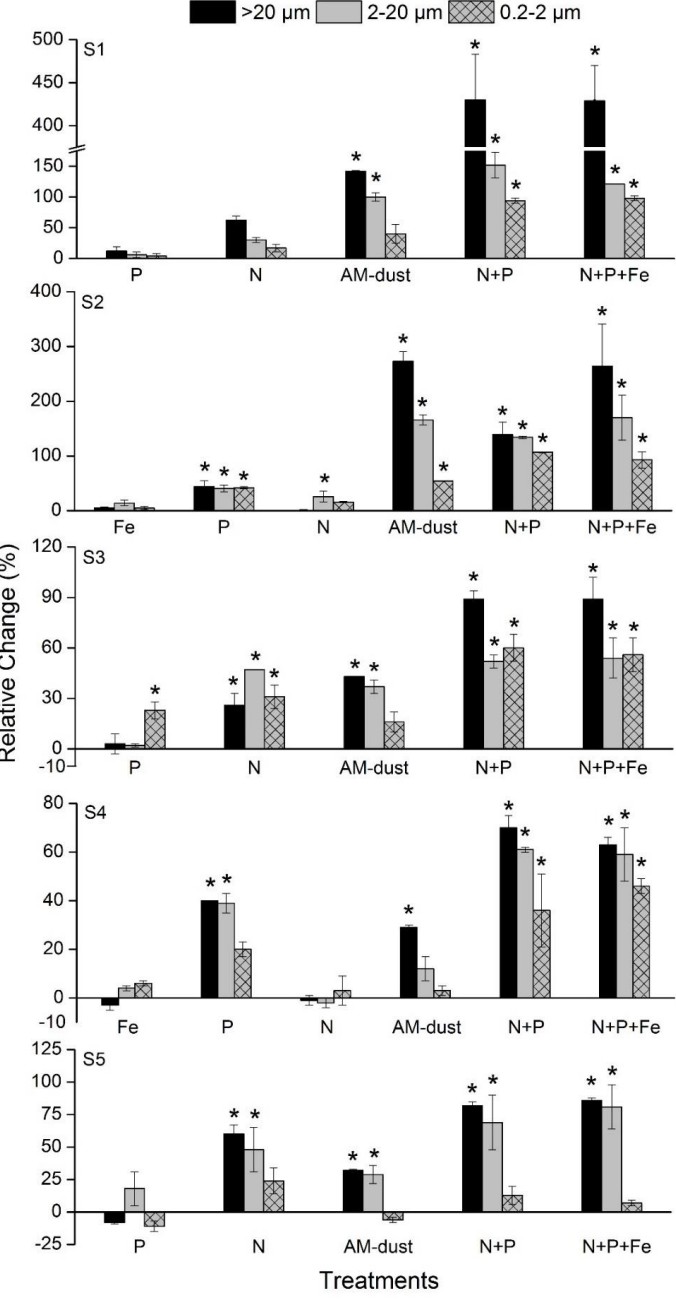

**Figure 7.** Relative change (%) in size-fractionated Chl *a* in AM-dust and various nutrient treatments during the successive increase in incubation period at each station. Relative change in this study was calculated as $100 \times$ ([mean in treatments - mean in control]/mean in control). Error bars represent the standard deviation in parallel incubations. Asterisks indicate that the treatment induces a significant difference in the mean of Chl *a* compared with those in control ($p < 0.05$).






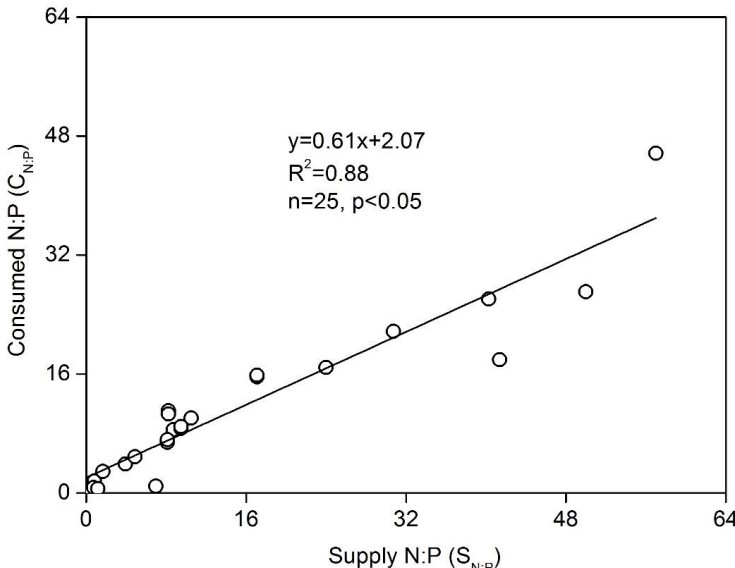

**Figure 8.** The relationship between the consumed N:P ratio ($C_{N:P}$) and supply N:P ratio ($S_{N:P}$) in the control and the various nutrient treatments during the successive increase in the incubation period at all stations.