# Peer review of "Phytoplankton growth responses to Asian dust additions in the Northwest Pacific Ocean versus the Yellow Sea"

_Biogeosciences, 2017_

## Referee Comment (RC1) · Anonymous Referee #1 · 14 Aug 2017

Review of Zhang et al. Biogeosciences

General comments

This study describes the results of several experiments in which surface plankton communities from the Yellow Sea and the NW Pacific ocean were amended with atmospheric dust and different nutrients added alone and in various combinations. The responses studied included the numerical abundance of diatoms and dinoflagellates, size-fractionated chl a concentration, and nutrient concentration. The main strength of the study is that parallel incubations, in which inorganic nutrients were added in different combinations, allowed the authors to gain insight into the causative mechanisms underlying the phytoplankton responses to dust. However, in some cases (as in the case of P availability, see below) the authors seem to over-interpret the available evidence. A limitation of the study is that only standing stocks were examined; no metabolic rate measurements were included, and therefore it is not possible to ascertain the dominant type of nutrient limitation (Blackman versus Liebig). The authors rely heavily on the use of chl a as a proxy for phytoplankton biomass. However, variability in C:Chla ratios should be taken into account. The extent to which the simulated process of atmospherical tranformation of dust yields materials that are realistic in terms of nutrient content and solubility should be discussed. Finally, some sections of the Discussion are speculative and based on tenuous assumptions. All these limitations should be addressed before publication is recommended in Biogeosciences. Some suggestions as to data presentation and analysis are also given below.

Specific comments

Simulation of atmospheric transformation of dust. How do these AM-dust materials compare, in terms of nutrient composition and solubility, with real dust samples collected in situ? This is critical to assess if the results observed are representative of real responses at sea. Table in supp. info. shows that N concentration is increased 4 orders of magnitude relative to N concentration in collected rain. Does this mean that the potential for nutrient supply is grossly overestimated in these artificially treated materials?

Section 3.2. This section should present first the changes in nutrient concentration, and then those of chla concentration. In both cases, the actual increases (absolute values) should be described (e.g. nutrient or chl a concentration increased by xx umol/L or ug/L), rather than just the relative increases (xx-fold). It is important to describe the chla responses in terms of absolute value of increase, so that they can be compared with the amount of nutrient released from the dust or provided by the nutrient amendments.

Conversion efficiency index. This index should be described in the Methods section.

It is unclear why the index is formulated in this way. Why not use just final minus initial chla concentration, as is done for N? It does not make sense to sum consecutive differences over time in chla concentration between treatments and control. In addition, the index has a potential flaw, because C:Chla ratios are bound to be different in the different sites (due, for instance, to differences in nutrient and/or light availability). So the same response in terms of % increase in biomass (carbon) will yield higher chla concentration (in absolute values), and thus higher conversion efficiency, in waters with low phytoplankton C:Chla values. The limitations of using Chla as a proxy for biomass should be acknowledged and discussed. Finally, when reporting the values of this index in the text, its units should be indicated.

Section 4.3. This section is speculative and difficult to follow. It is unclear how the 'increase in bioavailable P concentration following AM-dust addition' has been identified. The relationship between N:P ratios in supply vs demand is tentative at best, since actual supply N:P ratios were not measured. The paragraph on lines 395-406 starts with an untenable assumption, namely that 'C_N:P in AM-dust treatments is equal to that in N treatments'. To the extent that dust additions and N additions create distinct nutrient environments, it is most likely that consumption N:P ratios will be the same. In fact, the previous paragraph has argued that consumption N:P ratio is lower in dust treatments than in N treatments. Thus the subsequent calculations and conclusions have no use. This sub-section (l. 395-406) should be deleted. In the subsequent paragraph, the basis for the need for additional P supply is unclear.

Minor comments

lines 53-54: Rewrite sentence: 'The N:P ratio of dust deposition is much higher than the Redfield ratio (N:P=16)...'

line 110. Were attenuation filters used? PAR levels should be given.

line 139. The ultrasonic method should be described briefly. The use of ultrasounds maximises the extraction of nutrients but it probably overestimates the amount of nutrients that is actually released in real conditions at sea.

line 141. Re-write sentence, '...and filtrates were stored...'

line 153: '...enumeration of...

line 171: delete 'evidently'.

line 174: 'trophic level' means something else. Replace by appropriate phrase.

line 185: Why is P gained during the treatment?

line 189 Here cite Fig. 3, otherwise the reader does not know where is that increase reported. Tables do not report the nutrient increases observed in the treatments.

line 194 and elsewhere (including Fig. legends). The phrase 'successive increase' should be omitted. Sentence should read simply: 'During the incubation...'

line 271. Remove 'certain amount of'

line 300. This sentence seems to assume that all N present in the dust becomes bioavailable, because the concentrations referred to are those given in Table 3 (which correspond to concentrations in the dust, not in seawater).

line 448-449. Here the authors are deriving biogeochemical conclusions on the functioning of the biological pump, but their data consider just phytoplankton. Without information on how the metabolic activity of heterotrophs, bacteria in particular, changes in response to nutrient/dust additions, the ultimate effect on the biological pump remains unknown.

Table 1. Silicate measurements are missing – they would have been helpful to constrain the stoichiometry of diatom blooms in response to nutrient/dust amendments

Table 3. The data labelled 'increased concentrations' are theoretical or expected concentrations, assuming 100% of the nutrients in the dust becomes dissolved. This should be explicitly acknowledged in the Table legend. Is there any evidence to support the tenet that, upon dust deposition onto the ocean's surface, all nutrients become dissolved and bioavailable?

Figure legends: The phrase 'successive increase' is awkward. Delete in all fig. legends. It should be simply: 'changes in xxxx during the incubation period at each station'.

Fig. 2. Y-axis intervals should be regular (e.g., 0.5 or 1.0 ug/L) and consistent in all plots. Minor ticks should be included, to help the reader ascertain the magnitude of responses.

Fig. 3. Symbol for control should be more visible (it is often masked by other symbols).

Fig. 5. Revise species names spelling (e.g. Skeletonema). Species names and genera should be written in italics (but not 'spp.').

Figs. 6 and 7. The index values shown here result from the subtraction and division of variables measured independently, each with its own error. Therefore the error bars shown should be computed using the error propagation formulae for addition and division.

Fig. 8. These N:P ratios should be defined in the Methods section. Strictly speaking, the N:P supply ratio is not known, since no solubility experiments were conducted.

---

## Referee Comment (RC2) · Anonymous Referee #3 · 21 Aug 2017

Title: Phytoplankton growth responses to Asian dust additions in the Northwest Pacific Ocean versus Yellow Sea

Authors: Chao Zhang, Huiwang Gao, Xiaohong Yao, Zongbo Shi, Jinhui Shi, Yang Yu, Ling Meng, and Xinyu Guo

Journal: Biogeosciences

General Comments:

Biological productivity of the open ocean regions, especially oligotrophic parts, have attracted the attention of several researchers. Our present understanding of the role of

mineral dust in enhancing the primary productivity of the oligotrophic ocean by supplying bio-available nutrients is still in its infancy. The present manuscript aims at advancing such an understanding of enhancement of phytoplankton growth and the phytoplankton community structure by the nutrient supply from the Asian in the subtropical gyre, Kuroshio Extension of Northwest Pacifica Ocean and Yellow Sea region. The authors attempt this by incubation experiment onboard R/V Dongfanghong II during spring. For incubation experiment, the mineral dust collected from Gobi Desert region was artificially modified and phytoplankton assemblages from subtropical gyre, Northwest Pacific ocean and Yellow Sea region were used. Each of the five microcosm experiment lasted for 9 to 10 days. Using a net conversion efficiency index, proposed by the authors, of nitrogen conversion to chlorophyll a the authors explore the role of bio-available nutrients from mineral dust in the primary production at the above mentioned three regions.

The subject matter of the manuscript addresses an important aspect of phytoplankton growth by "bio-available nutrients" from "treated soil from Gobi desert" which is "expected to" simulate the natural mineral dust. Though there are several concerns that have been listed under specific comments, in my opinion, the results are publishable; but only after adequately addressing the concerns.

Major concerns:

1. The major concern is that the authors have not succeeded in unambiguously resolving the issue of quantification of bioavailability of nutrients form artificially modified mineral dust, especially phosphorous, for phytoplankton growth, which is the central theme of the manuscript. See for example, lines 334-341. The ambiguity regarding the "missing N and P".

2. How would the authors differentiate the phytoplankton growth-response due to N/P/Fe and that due to Mn and Zn (see for example, Saito et al., 2008; Sunda 2012).

3. The authors do not observe any community shift in the phytoplankton in their study.

[Figure]

There is no discussion on this aspect and authors need to address this.

4. Based on the information provided, it is hard to see how closely the artificially modified soil collected from Gobi desert mimics the nature. Some more robust information on the atmospheric (chemical) processing of the mineral dust during the long-range transport from source to the proposed study site during spring is needed along with the upper air wind vectors. What are the chemical constituents of such atmospheric processed mineral dust in presence of anthropogenic aerosol is not clear.

5. It is not clear from the manuscript, whether the ocean atmospheric conditions during the 3-month period (March-May 2014) at each of the sampling locations could be considered as a part of the same season where ocean and atmosphere represents similar conditions. The data from the Table 1 do not support this. For example, the average temperature (is it SST?) at S4 is quite different from the rest of the stations, which was sampled in May 2014. Similar, the MLD also is quite different. The authors need to address these issues.

6. The authors need to discuss the efficacy of the proposed "net conversion efficiency index" in varying Redfield ratio conditions.

7. Most of the results obtained from the incubation experiment, such as co-limitation of nutrients, response of phytoplankton biomass and structure, are largely known as could be seen from the literature cited in the manuscript. For example, co-limitation in the south China Sea and its response to aeolian input(Gao et al., 2012), Fu et al. (2009) on N:P ratios during spring, Fu et al (2009) study on the phytoplankton biomass and structure in South China Sea. Nishibe et al. (2015) work in the Kuroshio Extension during spring. See also under Minor comments (8).

Minor concerns:

8. Lines 310-311: Authors need to at least briefly state what are those "Complex hydrographic conditions"

9. Lines 339-342: This is purely speculative and needs further substantiation.

10. Lines 307-308: So what is new/different from the work of Nishibe et al. (2015).

11. Line 143: Expand SPSS

12. Table 3 : 2ns foot note (b) is missing in the Table

13. Also explain "E-3, E-4"

---

## Referee Comment (RC3) · Anonymous Referee #2 · 24 Aug 2017

Review of "Phytoplankton growth responses to Asian dust additions in the Northwest Pacific Ocean versus the Yellow Sea" by Chao Zhang et al.

General comments: This paper present a set of microcosm experiments performed on-board using sea water collected from two distinct oceanic regime: 1) Oligotrophic waters (Northwest Pacific Ocean) and 2) nutrient rich waters (Yellow Sea), to understand the impact of atmospheric dust (or processed dust) and other nutrients (N, P, Fe e.t.c) on the phytoplankton productivity in terms of increase in chlorophyll a (Chl a) and abundance of phytoplankton in various size fractions (e.g. micro, nano, pico e.t.c). The artificially modified atmospheric dust (AM-dust) is prepared using surface soil collected

from Gobi Desert. The set of experiments lasted for 9-10 days and clearly indicate an overall increase in Chl a concentrations due to addition of various combination of nutrients including AM-dust. Authors have proposed a "new" net conversion efficiency index (NCEI) to better understand the impact of specific nutrients (N, P, Fe, and AM-dust) on primary productivity at the sampled locations. The presented case study makes an important contribution towards improving our understanding on impact of aeolian deposition (external source of nutrients) on productivity and ocean biogeochemistry. The results obtained from set of experiments are discussed well, manuscript is easy to read (except few sections, see in specific comments) and should be of great interest to the Biogeoscience community. So, I recommend this paper for publication in Biogeosciences, but after addressing some of the concerns detailed in specific comments.

Specific comments: 1) Section 2.1: What is the size distribution of soil samples used for preparing AM-dust? This information is important because if, majority of collected soil particle are in coarse fraction (e.g. more than 30 microns), most of them gets deposited at the source region and hardly get transported to the Pacific. So, the soil used for AM-dust preparation is not at all a representative undergoing long-range transport and depositing on surface waters. The fine fraction (less than 5 microns) or typically clay fraction of the soil is a more representative dust which can be artificially processed to mimic the processed aeolian dust.

2) Section 2.2: Line 110-112: Why Day 1 was not sampled?

3) Section 2.4: The ultrasonic bath treatment may overestimate the nutrient concentration. What was the time duration used for ultra-sonication? More than 30 minute of treatment will increase the temperature and may enhance the leaching of nutrients and thus overestimate. Usually, the treatment is done for aerosol samples collected on filter substrate to loosen the particles from matrix.

4) Section 4.1: Line 328-329: This is a speculative statement and of course, it need further investigation. Except here, the role of trace metals (as nutrient or toxicant)

[Figure]

is nowhere discussed. The concentration of suite of trace metals (in AM-dust) are very low except Fe and Mn. This once again indicate that, AM-dust is not the best representative for processed dust.

5) Section 4.2: It is very difficult to follow the proposed conversion efficiency index (NCEI). It may be a good tool to specifically understand the role of nutrients on nitrogen consumption or productivity, but need to be elaborated more. It is not clear, why summation of differences of treatment and control for consecutive days are used?

6) Section 4.3: Line 408-419: This paragraph is mostly speculative and difficult to follow, although authors have concluded the importance of DOP determination in seawater.

Minor comments:

Line 181: should be Table 1.

The legends used in Fig. 2, 3 and 4 for nutrients (other than AM-dust) are in same colors and very hard to make out. Most of them are superimposed. It will be useful for reader if different coloured legends with connecting lines can be used.

---

## Author Comment (AC1) · 21 Sep 2017

**Referee #1**

*General comments:*

*This study describes the results of several experiments in which surface plankton communities from the Yellow Sea and the NW Pacific ocean were amended with atmospheric dust and different nutrients added alone and in various combinations. The responses studied included the numerical abundance of diatoms and dinoflagellates, size-fractionated chl a concentration, and nutrient concentration. The main strength of the study is that parallel incubations, in which inorganic nutrients were added in different combinations, allowed the authors to gain insight into the causative mechanisms underlying the phytoplankton responses to dust. However, in some cases (as in the case of P availability, see below) the authors seem to over-interpret the available evidence. A limitation of the study is that only standing stocks were examined; no metabolic rate measurements were included, and therefore it is not possible to ascertain the dominant type of nutrient limitation (Blackman versus Liebig). The authors rely heavily on the use of chl a as a proxy for phytoplankton biomass. However, variability in C:Chla ratios should be taken into account. The extent to which the simulated process of atmospherical tranformation of dust yields materials that are realistic in terms of nutrient content and solubility should be discussed. Finally, some sections of the Discussion are speculative and based on tenuous assumptions. All these limitations should be addressed before publication is recommended in Biogeosciences. Some suggestions as to data presentation and analysis are also given below.*

Response:

We would like to thank the referee very much for the valuable comments which enabled us to improve the quality of the manuscript. We will revise the manuscript accordingly to address the comments.

Regarding concern over nutrient limitation status: The standing stocks such as Chl *a* and biomass have been widely used for ascertaining the dominant type of nutrient limitation in the ocean. However, indeed, we acknowledge that

metabolic rate has a better representation than standing stocks in ascertaining nutrient limitation. We will add metabolic rate measurements for future incubation experiments.

Regarding C;Chl *a* ratios, we will illustrate that our results are based on Chl *a* and cell abundance rather than biomass, and also consider the uncertainty in the new manuscript (refer to Q4).

We will provide additional descriptions about the simulating processes in Text S1 (refer to Q1).

Regarding concerns over P, we have made a detailed response to Q5.

*Specific comments*

*1. Simulation of atmospheric transformation of dust. How do these AM-dust materials compare, in terms of nutrient composition and solubility, with real dust samples collected in situ? This is critical to assess if the results observed are representative of real responses at sea. Table in supp. info. shows that N concentration is increased 4 orders of magnitude relative to N concentration in collected rain. Does this mean that the potential for nutrient supply is grossly overestimated in these artificially treated materials?*
Response:
During a strong Asian dust event, the loading of inorganic nitrogen ($NO_3^-+NH_4^+$), soluble P and Fe in atmospheric particles collected in the Yellow Sea was around 714 µmol g$^{-1}$, 4.3 µmol g$^{-1}$ and 550 µg g$^{-1}$, respectively (Shi et al., 2012). The values in AM-dust used in this study are 577 µmol g$^{-1}$, 4.3 µmol g$^{-1}$ and 473 µg g$^{-1}$, respectively (Table 3 in the origin version), which were highly comparable to those observed by Shi et al. (2012). The authors, however, agree that the loadings of these components in atmospheric dust particles could highly vary in different cases. The weakness will be added at the end of the conclusion.

In this study, the aging process of dust followed Guieu's (2010) method and aimed at stimulating the cloud reaction between dust and synthetic evaporating cloud water. The pH around dust in the cloud process (i.e. mix with evaporating cloud water) was found to be as low as ~1 during their transport to the Yellow Sea (Meskhidze et al., 2003), whereas the typical pH in rainwater is 5 (Watanabe et al. 2001, Sasakawa and Uematsu, 2002, Wang et al. 2002, Sakihama et al. 2008, Zhang et al. 2011), meaning that a dilution by a factor $10e^4$. In consequent, in order to reproduce an evaporating cloud, we have used a concentration 10 000 in our experiments in comparison to the typical concentrations found in rainwater. Considering the typical concentrations of dust in rainwaters was 10 mg L$^{-1}$ (Ridame et al., 2002), the dust loading in evaporating cloud water could reach 100 g L$^{-1}$. As a consequence, all of the concentrations in evaporating cloud water were around 10000-fold larger (i.e. 4 orders of magnitude larger) than those in natural rainwater. Table S1 summarized the primary chemical composition of rains in the Eastern Asian regions and the evaporating cloud water used for our simulation.

Shi, J., Gao, H., Zhang, J., Tan, S., Ren, J., Liu, C., Liu, Y., and Yao, X.: Examination of causative link between a spring bloom and dry/wet deposition of Asian dust in the Yellow Sea, China, Journal of Geophysical Research: Atmospheres, 117, 2012, doi:10.1029/2012JD017983.

Guieu, C., Dulac, F., Desboeufs, K., Wagener, T., Pulido-Villena, E., Grisoni, J.-M., Louis, F., Ridame, C., Blain, S., and Brunet, C.: Large clean mesocosms and simulated dust deposition: a new methodology to investigate responses of marine oligotrophic ecosystems to atmospheric inputs, Biogeosciences, 7, 2765-2784, doi: 10.5194/bg-7-2765-2010, 2010.

Meskhidze, N., Chameides, W. L., Nenes, A., and Chen, G.: Iron mobilization in mineral dust: Can anthropogenic SO2 emissions affect ocean productivity?. Geophysical Research Letters, 30(21), 2003, doi: 10.1029/2003GL018035.

Watanabe, K., Ishizaka, Y., & Takenaka, C.: Chemical characteristics of cloud water over the Japan Sea and the Northwestern Pacific Ocean near the central part of Japan: airborne measurements. Atmospheric Environment, 35(4), 645-655, 2001, doi: 10.1016/S1352-2310(00)00358-7.

Sasakawa, M., & Uematsu, M.: Chemical composition of aerosol, sea fog, and rainwater in the marine boundary layer of the northwestern North Pacific and its marginal seas. Journal of Geophysical Research: Atmospheres, 107(D24), 2002, doi: 10.1029/2001JD001004.

Wang, Z., Akimoto, H., & Uno, I.: Neutralization of soil aerosol and its impact on the distribution of acid rain over east Asia: Observations and model results. Journal of Geophysical Research: Atmospheres, 107(D19), 2002, doi: 10.1029/2001JD001040.

Sakihama, H., Ishiki, M., & Tokuyama, A.: Chemical characteristics of precipitation in Okinawa Island, Japan. Atmospheric Environment, 42(10), 2320-2335, 2008, doi:    10.1016/j.atmosenv.2007.12.026.

Zhang, J., Zhang, G. S., Bi, Y. F., & Liu, S. M.: Nitrogen species in rainwater and aerosols of the Yellow and East China seas: Effects of the East Asian monsoon and anthropogenic emissions and relevance for the NW Pacific Ocean. Global Biogeochemical Cycles, 25(3), 2011, doi: 10.1029/2010GB003896.

Ridame, C., & Guieu, C.: Saharan input of phosphate to the oligotrophic water of the open western Mediterranean Sea. Limnology and Oceanography, 47(3), 856-869, 2002, doi: 10.4319/lo.2002.47.3.0856.

*2. Section 3.2. This section should present first the changes in nutrient concentration, and then those of chla concentration. In both cases, the actual increases (absolute values) should be described (e.g. nutrient or chl a concentration increased by xx umol/L or ug/L), rather than just the relative increases (xx-fold). It is important to describe the chla responses in terms of absolute value of increase, so that they can be compared with the amount of nutrient released from the dust or provided by the nutrient amendments.*

Response:

Agree. We will revise accordingly through the manuscript.

*3. Conversion efficiency index. This index should be described in the Methods section.*

Response:
Agree. It will be moved to the Method.

*4. It is unclear why the index is formulated in this way. Why not use just final minus initial chla concentration, as is done for N? It does not make sense to sum consecutive differences over time in chla concentration between treatments and control. In addition, the index has a potential flaw, because C:Chla ratios are bound to be different in the different sites (due, for instance, to differences in nutrient and/or light availability). So the same response in terms of % increase in biomass (carbon) will yield higher chla concentration (in absolute values), and thus higher conversion efficiency, in waters with low phytoplankton C:Chla values. The limitations of using Chla as a proxy for biomass should be acknowledged and discussed. Finally, when reporting the values of this index in the text, its units should be indicated.*

Response:

In the revision, we will clarify the net conversion efficiency index (NCEI) proposed in this study to be an approximate estimation for the utilization of N for the growth of phytoplankton. Therefore, the capacity to synthesize Chl *a* per unit concentration of nitrogen (N) in different treatments can be compared. We agree that the sum consecutive differences over time in Chl *a* concentration between treatments and control will lead to an overestimation of the real net conversion efficiency because of the accumulation effect. Theoretically, the use of the maximum difference will lead to an underestimation of the real net conversion efficiency because of degradation of Chl *a* in the growth period. The real net conversion efficiency should be between those calculated by the two approaches.

We agree that C:Chl *a* ratios can be different at the different sites. However, most of the calculated indexes at different sites in this study were highly consistent. This implied that C:Chl *a* ratios at these site in this study might narrowly vary. However, it should be cautious for extrapolating the current findings to other sites. This will be clarified in the revision.

The unit of NCEI in the text will be added in the revision.

*5. Section 4.3. This section is speculative and difficult to follow. It is unclear how the 'increase in bioavailable P concentration following AM-dust addition' has been identified. The relationship between N:P ratios in supply vs demand is tentative at best, since actual supply N:P ratios were not measured. The paragraph on lines 395-406 starts with an untenable assumption, namely that 'C_N:P in AM-dust treatments is equal to that in N treatments'. To the extent that dust additions and N additions create distinct nutrient environments, it is most unlikely that consumption N:P ratios will be the same. In fact, the previous paragraph has argued that consumption N:P ratio is lower in dust treatments than in N treatments. Thus the subsequent calculations and conclusions have no use. This sub-section (l. 395-406) should be deleted. In the subsequent paragraph, the basis for the need for additional P supply is unclear.*

Response:

We have determined the chemical contents of AM-dust in the laboratory. The details can be seen in Methods Section 2.4. The content of bioavailable nutrients from AM-dust has been listed in Table 3. As the added concentration of AM-dust was 2 mg L$^{-1}$, thus, we can get the theoretical amounts of N and P nutrients supplied by AM-dust in the incubation system. In the field incubation experiments, the time interval of adding materials (e.g. AM-dust and various nutrients) into the incubation bottles and nutrient measurement (i.e. measurement on day 0) was around 1-2h. During this period, microbial uptake, scavenging by cell surface and bottle wall are all possible to influence the measurement of added N and P nutrients from AM-dust, which has been proved by the previous incubation experiments (Liu et al., 2013). Therefore, the calculated theoretical amounts of N and P nutrients supplied by AM-dust in the incubation system is close to the actual values relative to those measured directly on day 0.

We will correct the inappropriate expression of '$C_{N:P}$ in AM-dust treatments is equal to that in N treatments' and change the equation (2) into inequation (2).

The last paragraph in the Discussion Section 4.3 is particularly important because it provides a new insight to analyze the budget of bioavailable P in the AM-dust addition incubation experiments. Unfortunately, the original presentation seemed to be unclear and didn't service the target well. We will rewrite this paragraph to make our thoughts understandable.

Liu, Y., Zhang, T., Shi, J., Gao, H., and Yao, X.: Responses of chlorophyll a to added nutrients, Asian dust, and rainwater in an oligotrophic zone of the Yellow Sea: Implications for promotion and inhibition effects in an incubation experiment, Journal of Geophysical Research: Biogeosciences, 118, 1763-1772, doi: 10.1002/2013JG002329, 2013.

Minor comments

*1. lines 53-54: Rewrite sentence: 'The N:P ratio of dust deposition is much higher than the Redfield ratio (N:P=16)...'*
Response:
Corrected.

*2. line 110. Were attenuation filters used? PAR levels should be given.*
Response:
We did not use attenuation filters in this study. The surface seawater (2-5 m) was collected and incubated under natural light.

*3. line 139. The ultrasonic method should be described briefly. The use of ultrasounds maximises the extraction of*

*nutrients but it probably overestimates the amount of nutrients that is actually released in real conditions at sea.*

Response:

We will make a brief description of the ultrasonic method in the text.

In general, more than 30 minute of ultra-sonication treatment will increase the temperature and may enhance the leaching of nutrients and thus overestimate. In our study, the time duration used for ultra-sonication is 30 minute. We used the ice pack to keep the temperature of water bath stable at ~0℃. Please also see Specific Comments No. 3 for Referee #3.

*4. line 141. Re-write sentence, '...and filtrates were stored...'*

Response:

Corrected.

*5. line 153: '...enumeration of...*

Response:

Corrected.

*6. line 171: delete 'evidently'.*

Response:

Corrected.

*7. line 174: 'trophic level' means something else. Replace by appropriate phrase.*

Response:

We will replace 'trophic level' with 'trophic state'.

*8. line 185: Why is P gained during the treatment?*

Response:

The acid process can enhance the solubility of P in the mineral dust.

*9. line 189 Here cite Fig. 3, otherwise the reader does not know where is that increase reported. Tables do not report the nutrient increases observed in the treatments.*

Response:

Corrected.

*10. line 194 and elsewhere (including Fig. legends). The phrase 'successive increase' should be omitted. Sentence should read simply: 'During the incubation...'*

Response:

The duration of the incubation experiment in this study is 9~10 days, but we mainly analyzed the data on the initial 3-5 days, in which the Chl *a* concentration showed a successive increase. Thus, we used the 'successive increase' to distinguish the initial 3-5 days from the whole incubation period (9-10 days).

*11. line 271. Remove 'certain amount of'.*
Response:
Corrected.

*12. line 300. This sentence seems to assume that all N present in the dust becomes bioavailable, because the concentrations referred to are those given in Table 3 (which correspond to concentrations in the dust, not in seawater).*
Response:
We will rewrite the sentence. The contents of N and P in the untreated and AM-dust listed in Table 3 were determined in seawater. The details can be seen in the description of 'ultrasonic bath'.

*13. line 448-449. Here the authors are deriving biogeochemical conclusions on the functioning of the biological pump, but their data consider just phytoplankton. Without information on how the metabolic activity of heterotrophs, bacteria in particular, changes in response to nutrient/dust additions, the ultimate effect on the biological pump remains unknown.*
Response:
Thank you for your thoughtful suggestion. We will rewrite the sentence and illustrate that the enhanced biological pump was only a possibility.

*14. Table 1. Silicate measurements are missing – they would have been helpful to constrain the stoichiometry of diatom blooms in response to nutrient/dust amendments.*
Response:
In the original experimental settings, we aimed at exploring the relationship between phytoplankton growth and supplied bioavailable nutrients (N, P, and Fe) by AM-dust additions. Thus, we did not determine silicate (Si) concentrations. For the Yellow Sea, Si is generally not a limiting nutrient for the growth of phytoplankton in spring because of the influence of riverine input (Wang et al., 2003). Si may become a limiting nutrient during the diatom blooms in the open ocean of the northwest Pacific, but it will not influence the result of this study, which was mainly concerned with N, P, and Fe nutrients. We thank the comment and will add the measurement of silicate for future incubation experiments.

Wang, B. D., Wang, X. L., and Zhan, R. Nutrient conditions in the Yellow Sea and the East China Sea. Estuarine, Coastal and Shelf Science, 58, 127-136, doi: 10.1016/S0272-7714(03)00067-2, 2003.

*15. Table 3. The data labelled 'increased concentrations' are theoretical or expected concentrations, assuming 100% of the nutrients in the dust becomes dissolved. This should be explicitly acknowledged in the Table legend. Is there any evidence to support the tenet that, upon dust deposition onto the ocean's surface, all nutrients become dissolved and bioavailable?*
Response:
We will add 'theoretically' in the Table legend.

The $NO_3^-+NO_2^-$ in the AM-dust would dissolved almost thoroughly once exposing to the filtered seawater (Ridame et al., 2014) while the dissolution of $PO_4^{3-}$ would continue over multiple days because the dissolution of less labile (but still soluble) P compounds would take some time to dissolve in seawater (Mackey et al., 2012). But phytoplankton in the incubated bottles can absorb all the soluble $NO_3^-+NO_2^-$ and $PO_4^{3-}$ from AM-dust if they need.

Ridame, C., Dekaezemacker, J., Guieu, C., Bonnet, S., L'Helguen, S., and Malien, F.: Contrasted Saharan dust events in LNLC environments: impact on nutrient dynamics and primary production, Biogeosciences, 11, 4783-4800, doi: 10.5194/bg-11-4783-2014, 2014.
Mackey, K. R., Roberts, K., Lomas, M. W., Saito, M. A., Post, A. F., and Paytan, A.: Enhanced solubility and ecological impact of atmospheric phosphorus deposition upon extended seawater exposure, Environmental science & technology, 46, 10438-10446, doi: 10.1021/es3007996, 2012.

*16. Figure legends: The phrase 'successive increase' is awkward. Delete in all fig. legends. It should be simply: 'changes in xxxx during the incubation period at each station'.*
Response:
Please see 'Minor comments-No.10.

*17. Fig. 2. Y-axis intervals should be regular (e.g., 0.5 or 1.0 ug/L) and consistent in all plots. Minor ticks should be included, to help the reader ascertain the magnitude of responses.*
Response:
Corrected.

*18. Fig. 3. Symbol for control should be more visible (it is often masked by other symbols).*
Response:
Corrected.

*19. Fig. 5. Revise species names spelling (e.g. Skeletonema). Species names and genera should be written in italics (but not 'spp.').*
Response:
Corrected.

*20. Figs. 6 and 7. The index values shown here result from the subtraction and division of variables measured independently, each with its own error. Therefore the error bars shown should be computed using the error propagation formulae for addition and division.*

Response:

Thanks for the valuable suggestion. We will use the error propagation formulae to calculate the error bars.

*21. Fig. 8. These N:P ratios should be defined in the Methods section. Strictly speaking, the N:P supply ratio is not known, since no solubility experiments were conducted.*

Response:

We will move the definitions of N:P ratios to the Methods section. The dissolved concentrations of N and P from AM-dust in the seawater have been determined in the laboratory (Table 3). Please also see 'Specific comments-No.5.

**Referee #2**

*General comments*

*1. Biological productivity of the open ocean regions, especially oligotrophic parts, have attracted the attention of several researchers. Our present understanding of the role of mineral dust in enhancing the primary productivity of the oligotrophic ocean by supplying bio-available nutrients is still in its infancy. The present manuscript aims at advancing such an understanding of enhancement of phytoplankton growth and the phytoplankton community structure by the nutrient supply from the Asian in the subtropical gyre, Kuroshio Extension of Northwest Pacifica Ocean and Yellow Sea region. The authors attempt this by incubation experiment onboard R/V Dongfanghong II during spring. For incubation experiment, the mineral dust collected from Gobi Desert region was artificially modified and phytoplankton assemblages from subtropical gyre, Northwest Pacific ocean and Yellow Sea region were used. Each of the five microcosm experiment lasted for 9 to 10 days. Using a net conversion efficiency index, proposed by the authors, of nitrogen conversion to chlorophyll a the authors explore the role of bio-available nutrients from mineral dust in the primary production at the above mentioned three regions.*

*2. The subject matter of the manuscript addresses an important aspect of phytoplankton growth by "bio-available nutrients" from "treated soil from Gobi desert" which is "expected to" simulate the natural mineral dust. Though there are several concerns that have been listed under specific comments, in my opinion, the results are publishable; but only after adequately addressing the concerns.*

Response:

We greatly appreciate the referee for the thoughtful comments and will revise our manuscript accordingly.

Major concerns:

*1. The major concern is that the authors have not succeeded in unambiguously resolving the issue of quantification of bioavailability of nutrients form artificially modified mineral dust, especially phosphorous, for phytoplankton growth, which is the central theme of the manuscript. See for example, lines 334-341. The ambiguity regarding the 'missing N and P'.*

Response:

We are sorry that the word of 'missing' in the manuscript confused the referee. The 'missing N and P' and 'missing parts' in the text meant the N and P might be adsorbed on the bottle walls, suspended particles, and phytoplankton in the solution, while 'missing sources' meant there might be other sources to supply bioavailable P. Thus, we will change 'missing sources' to 'other sources' to express them clearly. Considering the comment and the similar comments from other two referees, we will rewrite the part related to the budget of P in incubation experiments. Please see our revised manuscript.

*2. How would the authors differentiate the phytoplankton growth-response due to N/P/Fe and that due to Mn and Zn (see for example, Saito et al., 2008; Sunda 2012).*

Response:

We used the various nutrient addition experiments to illustrate which nutrients (i.e. N, P, and Fe) limit phytoplankton growth. As we did not conduct Mn or Zn addition incubation experiments and determine the concentration of Mn and Zn in the seawater, it is difficult for us to accurately see whether phytoplankton were under limitation of Mn and/or Zn. However, through making a comparison of the net conversion efficiency index (NCEI) in various nutrient addition treatments, we can roughly determine whether there were other nutrients except N, P, and Fe dissolved from AM-dust to stimulate phytoplankton growth. For instance, the NCEI value in AM-dust treatment was significantly higher than that in N+P+Fe treatments, indicating the N, P, and Fe supplied by AM-dust cannot explain the corresponding phytoplankton growth. As reported in literature, trace metals such as Mn and Zn can affect phytoplankton growth rate (Sunda, 2012), which might lead to higher NCEI. The effect of Mn and Zn on phytoplankton growth is just a hypothesis, which will be clarified in the revision.

Sunda, W.: Feedback interactions between trace metal nutrients and phytoplankton in the ocean, Frontiers in microbiology, 3, 204, doi: 10.3389/fmicb.2012.00204, 2012.

*3. The authors do not observe any community shift in the phytoplankton in their study. There is no discussion on this aspect and authors need to address this.*

Response:

We thank the comment and will add some discussions about community shift of phytoplankton in Discussion

Section 4.2.

*4. Based on the information provided, it is hard to see how closely the artificially modified soil collected from Gobi desert mimics the nature. Some more robust information on the atmospheric (chemical) processing of the mineral dust during the long-range transport from source to the proposed study site during spring is needed along with the upper air wind vectors. What are the chemical constituents of such atmospheric processed mineral dust in presence of anthropogenic aerosol is not clear.*

Response:

During a strong Asian dust event, the loading of inorganic nitrogen ($NO_3^-$+$NH_4^+$), soluble P and Fe in atmospheric particles collected in the Yellow Sea was ~714 μmol $g^{-1}$, ~4.3 μmol $g^{-1}$ and ~550 μg $g^{-1}$, respectively (Shi et al., 2012). The values in AM-dust used in this study are 577 μmol $g^{-1}$, 4.3 μmol $g^{-1}$ and 473 μg $g^{-1}$, respectively (Table 3 in the origin version), which were highly comparable to those observed by Shi et al. (2012). The authors, however, agree that the loadings of these components in atmospheric dust particles could highly vary in different cases. The weakness will be added at the end of the conclusion.

We will add a description of preparing AM-dust in the Text S1:

In this study, the aging process of dust followed Guieu's (2010) method and aimed at stimulating the cloud reaction between dust and synthetic evaporating cloud water. The pH around dust in the cloud process (i.e. mix with evaporating cloud water) was found to be as low as ~1 during their transport to the Yellow Sea (Meskhidze et al., 2003), whereas the typical pH in rainwater is 5 (Watanabe et al. 2001, Sasakawa and Uematsu, 2002, Wang et al. 2002, Sakihama et al. 2008, Zhang et al. 2011), meaning that a dilution by a factor $10e^4$. In consequent, in order to reproduce an evaporating cloud, we have used a concentration 10 000 in our experiments in comparison to the typical concentrations found in rainwater. Considering the typical concentrations of dust in rainwaters was 10 mg $L^{-1}$ (Ridame et al., 2002), the dust loading in evaporating cloud water could reach 100 g $L^{-1}$. As a consequence, all of the concentrations in evaporating cloud water were around 10000-fold larger (i.e. 4 orders of magnitude larger) than those in natural rainwater. Table S1 summarized the primary chemical composition of rains in the Eastern Asian regions and the evaporating cloud water used for our simulation.

Shi, J., Gao, H., Zhang, J., Tan, S., Ren, J., Liu, C., Liu, Y., and Yao, X.: Examination of causative link between a spring bloom and dry/wet deposition of Asian dust in the Yellow Sea, China, Journal of Geophysical Research: Atmospheres, 117, 2012, doi:10.1029/2012JD017983.

Guieu, C., Dulac, F., Desboeufs, K., Wagener, T., Pulido-Villena, E., Grisoni, J.-M., Louis, F., Ridame, C., Blain, S., and Brunet, C.: Large clean mesocosms and simulated dust deposition: a new methodology to investigate responses of marine oligotrophic ecosystems to atmospheric inputs, Biogeosciences, 7, 2765-2784, doi: 10.5194/bg-7-2765-2010, 2010.

Meskhidze, N., Chameides, W. L., Nenes, A., and Chen, G.: Iron mobilization in mineral dust: Can anthropogenic

SO$_2$ emissions affect ocean productivity?. Geophysical Research Letters, 30(21), 2003, doi: 10.1029/2003GL018035.

Watanabe, K., Ishizaka, Y., & Takenaka, C.: Chemical characteristics of cloud water over the Japan Sea and the Northwestern Pacific Ocean near the central part of Japan: airborne measurements. Atmospheric Environment, 35(4), 645-655, 2001, doi: 10.1016/S1352-2310(00)00358-7.

Sasakawa, M., & Uematsu, M.: Chemical composition of aerosol, sea fog, and rainwater in the marine boundary layer of the northwestern North Pacific and its marginal seas. Journal of Geophysical Research: Atmospheres, 107(D24), 2002, doi: 10.1029/2001JD001004.

Wang, Z., Akimoto, H., & Uno, I.: Neutralization of soil aerosol and its impact on the distribution of acid rain over east Asia: Observations and model results. Journal of Geophysical Research: Atmospheres, 107(D19), 2002, doi: 10.1029/2001JD001040.

Sakihama, H., Ishiki, M., & Tokuyama, A.: Chemical characteristics of precipitation in Okinawa Island, Japan. Atmospheric Environment, 42(10), 2320-2335, 2008, doi: 10.1016/j.atmosenv.2007.12.026.

Zhang, J., Zhang, G. S., Bi, Y. F., & Liu, S. M.: Nitrogen species in rainwater and aerosols of the Yellow and East China seas: Effects of the East Asian monsoon and anthropogenic emissions and relevance for the NW Pacific Ocean. Global Biogeochemical Cycles, 25(3), 2011, doi: 10.1029/2010GB003896.

Ridame, C., & Guieu, C.: Saharan input of phosphate to the oligotrophic water of the open western Mediterranean Sea. Limnology and Oceanography, 47(3), 856-869, 2002, doi: 10.4319/lo.2002.47.3.0856.

*5. It is not clear from the manuscript, whether the ocean atmospheric conditions during the 3-month period (March-May 2014) at each of the sampling locations could be considered as a part of the same season where ocean and atmosphere represents similar conditions. The data from the Table 1 do not support this. For example, the average temperature (is it SST?) at S4 is quite different from the rest of the stations, which was sampled in May 2014. Similar, the MLD also is quite different. The authors need to address these issues.*

Response:

We are sorry that we made a mistake of sea surface temperature (SST) and mixed layer depth (MLD) at S5 and will correct them in the revision.

The seawater at S4 and S5 was sampled in May, which was later than the sampling date at other stations (Table 1). But the SST at both stations were lower than those at other stations. Thus the difference of SST between S4 and other stations was not caused by the different season. The S4 and S5 stations are located in the cold water regions, which will make SST in this region lower than that in the open oceans of the northwest Pacific.

The open ocean in the northwest Pacific is accompanied by complicated hydrological conditions such as frequent warm and cold eddies, which lead to strong water mixing in the upper ocean and may influence the MLD. While in the Yellow Sea, the seawater depth at S4 and S5 were lower than 100 m, which lead to a completely different MLD compared to those in the open oceans of the northwest Pacific.

We will add a description of SST and MLD conditions at the sampling locations in Results Section 3.1.

*6. The authors need to discuss the efficacy of the proposed "net conversion efficiency index" in varying Redfield ratio conditions.*
Response:
Good suggestion! We will add some discussions about the efficacy of the proposed "net conversion efficiency index" in varying Redfield ratio conditions in Discussion Section 4.2.

*7. Most of the results obtained from the incubation experiment, such as co-limitation of nutrients, response of phytoplankton biomass and structure, are largely known as could be seen from the literature cited in the manuscript. For example, co-limitation in the south China Sea and its response to aeolian input (Gao et al., 2012), Fu et al. (2009) on N:P ratios during spring, Fu et al (2009) study on the phytoplankton biomass and structure in South China Sea. Nishibe et al. (2015) work in the Kuroshio Extension during spring. See also under Minor comments (8).*
Response:
We recognize that previous studies have identified the nutrient limitation status in the study area. We are content that our nutrient addition results are consistent with theirs, as this confirms that the conditions we were studying were representative to the regions.

The similar results compared to other studies aimed at illustrating the following ideas:
The main strength of the study is to gain insight into the causative mechanisms underlying the phytoplankton responses to dust through parallel incubations, in which inorganic nutrients were added in different combinations. The added amount of inorganic nutrients (e.g. N: 2 $\mu$mol $L^{-1}$, P: 0.2 $\mu$mol $L^{-1}$, Fe: 2 nmol $L^{-1}$) was not equal to that dissolved from AM-dust, which leads to incomparable increases of Chl *a* concentrations in AM-dust and various nutrient treatments. Thus, we proposed the conversion efficiency index (NCEI) to quantify the role of N, P, and Fe dissolved from AM-dust played in stimulating phytoplankton growth, based on making a comparison with the results concluded from various nutrient addition treatments. Finally, we used the correlation of $S_{N:P}$ and $C_{N:P}$ to highlight increased bioavailability of P in AM-dust addition experiments.

Minor concerns:

*8. Lines 310-311: Authors need to at least briefly state what are those "Complex hydrographic conditions".*
Response:
We will add a brief description of complex hydrographic conditions in the revised manuscript.

*9. Lines 339-342: This is purely speculative and needs further substantiation.*

Response:

We will rewrite this sentence to express it clearly.

In the field incubation experiments, the time interval of adding materials (e.g. AM-dust and various nutrients) into the incubation bottles and nutrient measurement (i.e. measurement on day 0) was around 1-2h. During this period, microbial uptake, scavenging by cell surface and bottle wall are all possible to influence the measurement of added N and P nutrients from AM-dust, which lead to the concentrations determined on day 0 were lower than the theoretical adding amounts. When the concentrations of $NO_3^-+NO_2^-$ and $PO_4^{3-}$ decreased in the seawater, the $NO_3^-+NO_2^-$ and $PO_4^{3-}$ absorbed by cell surface and bottle wall had the potential to be released into the solution again for reaching equilibrium.

*10. Lines 307-308: So what is new/different from the work of Nishibe et al. (2015).*

Response:

Please see Major concerns No.7.

*11. Line 143: Expand SPSS*

Response:

Corrected.

*12. Table 3 : 2ns foot note (b) is missing in the Table*

Response:

Corrected.

*13. Also explain "E-3, E-4"*

Response:

We will add an explanation of 'E-3' and 'E-4' in Table 3.

**Referee #3**

*General comments*

*This paper present a set of microcosm experiments performed on-board using sea water collected from two distinct oceanic regime: 1) Oligotrophic waters (Northwest Pacific Ocean) and 2) nutrient rich waters (Yellow Sea), to*

*understand the impact of atmospheric dust (or processed dust) and other nutrients (N, P, Fe e.t.c) on the phytoplankton productivity in terms of increase in chlorophyll a (Chl a) and abundance of phytoplankton in various size fractions (e.g. micro, nano, pico e.t.c). The artificially modified atmospheric dust (AM-dust) is prepared using surface soil collected from Gobi Desert. The set of experiments lasted for 9-10 days and clearly indicate an overall increase in Chl a concentrations due to addition of various combination of nutrients including AM-dust. Authors have proposed a "new" net conversion efficiency index (NCEI) to better understand the impact of specific nutrients (N, P, Fe, and AM-dust) on primary productivity at the sampled locations. The presented case study makes an important contribution towards improving our understanding on impact of aeolian deposition (external source of nutrients) on productivity and ocean biogeochemistry. The results obtained from set of experiments are discussed well, manuscript is easy to read (except few sections, see in specific comments) and should be of great interest to the Biogeoscience community. So, I recommend this paper for publication in Biogeosciences, but after addressing some of the concerns detailed in specific comments.*

Response:

We greatly appreciate the referee for the constructive comments and will revise our manuscript accordingly.

*Specific comments:*

*1 Section 2.1: What is the size distribution of soil samples used for preparing AM-dust? This information is important because if, majority of collected soil particle are in coarse fraction (e.g. more than 30 microns), most of them gets deposited at the source region and hardly get transported to the Pacific. So, the soil used for AMdust preparation is not at all a representative undergoing long-range transport and depositing on surface waters. The fine fraction (less than 5 microns) or typically clay fraction of the soil is a more representative dust which can be artificially processed to mimic the processed aeolian dust.*

Response:

Recently, the transport routes of Asian dust move the northward. We had practical difficulty to collect the sufficient amount of ambient dust samples for incubation experiments. Alternatively, we used AM-dust for experiments as those reported by Guieu and Ridame, etc (Guieu et al., 2010; Ridame et al., 2014). In this study, we did not determine the size distribution of soil samples, but only the fraction less than 20 μm was used for preparing AM-dust in this study. Fe and P composition in $PM_{10}$ and $PM_{20}$ generated from the same soils were reported to be quite similar (Shi et al., 2011; Nenes et al., 2011). We agree that finer dust, e.g., less than 5 microns, should be more representative of those in the aeolian dust transported to the sea. However, it is also practically difficult to gain the sufficient amount of the fine dust for modification. Thus, it is a practical compromise by using artificially modified $PM_{20}$. The information will be added in the revision.

Guieu, C., Dulac, F., Desboeufs, K., Wagener, T., Pulido-Villena, E., Grisoni, J.-M., Louis, F., Ridame, C., Blain, S., and Brunet, C.: Large clean mesocosms and simulated dust deposition: a new methodology to investigate

responses of marine oligotrophic ecosystems to atmospheric inputs, Biogeosciences, 7, 2765-2784, doi: 10.5194/bg-7-2765-2010, 2010.

Ridame, C., Dekaezemacker, J., Guieu, C., Bonnet, S., L'Helguen, S., and Malien, F.: Contrasted Saharan dust events in LNLC environments: impact on nutrient dynamics and primary production, Biogeosciences, 11, 4783-4800, doi: 10.5194/bg-11-4783-2014, 2014.

Nenes, A., Krom, M. D., Mihalopoulos, N., Cappellen, P., Shi, Z., Bougiatioti, A., Zarmpas, P., and Herut, B.: Atmospheric acidification of mineral aerosols: a source of bioavailable phosphorus for the oceans, Atmospheric Chemistry and Physics, 11(13): 6265-6272, doi: 10.5194/acp-11-6265-2011, 2011.

Shi, Z., Bonneville, S., Krom, M. D., Carslaw, K.S., Jickells, T. D., Baker, A.R., and Benning L.G.: Iron dissolution kinetics of mineral dust at low pH during simulated atmospheric processing, Atmospheric Chemistry and Physics, 11(3): 995-1007, doi: 10.5194/acp-11-995-2011, 2011.

*2. Section 2.2: Line 110-112: Why Day 1 was not sampled?*

Response:

In our previous incubation experiments, there was no distinct difference of Chl *a* concentrations on day 1 and day 0 in all cases (Liu et al., 2013). Thus, we did not take a sample on day 1 in this study. We thank the comment and will change back our sampling protocol for future incubation experiments. We agree that the data on day 1 may be valuable. This will be clarified in the revision.

Liu, Y., Zhang, T., Shi, J., Gao, H., and Yao, X.: Responses of chlorophyll a to added nutrients, Asian dust, and rainwater in an oligotrophic zone of the Yellow Sea: Implications for promotion and inhibition effects in an incubation experiment, Journal of Geophysical Research: Biogeosciences, 118, 1763-1772, doi: 10.1002/2013JG002329, 2013.

*3 Section 2.4: The ultrasonic bath treatment may overestimate the nutrient concentration. What was the time duration used for ultra-sonication? More than 30 minute of treatment will increase the temperature and may enhance the leaching of nutrients and thus overestimate. Usually, the treatment is done for aerosol samples collected on filter substrate to loosen the particles from matrix.*

Response:

The time duration used for ultra-sonication is 30 minute. We used the ice pack to keep the temperature of water bath stable at ~0°C.

*4. Section 4.1: Line 328-329: This is a speculative statement and of course, it need further investigation. Except here, the role of trace metals (as nutrient or toxicant) is nowhere discussed. The concentration of suite of trace metals (in AM-dust) are very low except Fe and Mn. This once again indicate that, AM-dust is not the best representative for processed dust.*

Response:

The sentence will be revised as "Micro-nutrients, e.g., Mn, (Coale, 1991; Jakuba et al., 2008; Saito et al., 2008; Sunda, 2012) from the AM-dust may have contributed to the phytoplankton growth in the incubations at S2. This potential synergistic effect is worthy of further investigation."

We will also revise the part related to the role of trace metals (as nutrient or toxicant) other than Fe and Mn. As showed in our Table 3, most of trace metals are indeed negligible.

The aging process of dust in this study focused on the reaction between dust and inorganic acids ($H_2SO_4$ and $HNO_3$, details can be seen in Text S1). The distinguishing characteristic of AM-dust relative to untreated dust mainly reflects in the increased contents of soluble N, P, and trace metals. Trace metals Fe and Mn were mainly originated from mineral aerosols, thus we can observe noticeably enhanced solubility of Fe and Mn in the AM-dust relative to untreated dust. The contents of other soluble trace metals such as Cu, Pb, and Zn in the dust were mainly affected by anthropogenic factor such as automobile exhaust and coal combustion, which were not reflected well in our study. To this point, we will add an illustration at the end of the conclusion.

*5. Section 4.2: It is very difficult to follow the proposed conversion efficiency index (NCEI). It may be a good tool to specifically understand the role of nutrients on nitrogen consumption or productivity, but need to be elaborated more. It is not clear, why summation of differences of treatment and control for consecutive days are used?*
*The proposed conversion efficiency index (NCEI).*
Response:
In the revision, we will clarify the net conversion efficiency index (NCEI) proposed in this study to be an approximate estimation for the utilization of N for the growth of phytoplankton. Therefore, the capacity to synthesize Chl *a* per unit concentration of nitrogen (N) in different treatments can be compared. We agree that the sum consecutive differences over time in Chl *a* concentration between treatments and control will lead to an overestimation of the real net conversion efficiency because of the accumulation effect. Theoretically, the use of the maximum difference will lead to an underestimation of the real net conversion efficiency because of degradation of Chl *a* in the growth period. The real net conversion efficiency should be between those calculated by the two approaches.

We will add a more elaborated description of NCEI in Discussion Section 4.2.

*6. Section 4.3: Line 408-419: This paragraph is mostly speculative and difficult to follow, although authors have concluded the importance of DOP determination in seawater.*
Response: The part is particularly important because it provides a new insight to analyze the budget of bioavailable P in the AM-dust addition incubation experiments. Unfortunately, the original presentation seemed to be unclear and didn't service the target well. We will rewrite this paragraph to make our thoughts understandable.

Minor comments:

*7. Line 181: should be Table 1.*
Response:
Corrected.

*8. The legends used in Fig. 2, 3 and 4 for nutrients (other than AM-dust) are in same colors and very hard to make out. Most of them are superimposed. It will be useful for reader if different coloured legends with connecting lines can be used.*
Response:
Corrected.

---

## Author Comment (AC2) · 6 Nov 2017

**Additional changes we made to improve the presentation:**

Line 52: Changed 'worthy' to 'worthy of';

Line 163: Changed 'Selection of study period' to 'Protocol of data analysis';

Line 200-201: Changed 'For the S5 station in the YS, although the nutrient levels were comparable to those at S1, Chl *a* concentrations were as high as 2.74 μg L$^{-1}$, close to the conditions of spring bloom in the YS' to 'Although the nutrient levels at S5 in the YS were comparable to those at S1, the Chl *a* concentration of 2.74 μg L$^{-1}$ was close to the values during spring blooms in the YS';

Line 204-205: Changed 'S1 and S2 (< 20 cells mL$^{-1}$) to 79 cells mL$^{-1}$ at S3 and 35 cells mL$^{-1}$ at S4' to '< 20 cells mL$^{-1}$ at S1 and S2 to 35 cells mL$^{-1}$ at S4 and 79 cells mL$^{-1}$ at S3';

Line 209-210: Changed 'As shown in Table 3, the concentration of dissolved inorganic nitrogen (DIN, i.e. $NO_3^-+NO_2^-+NH_4^+$) in the AM-dust was 577 μmol g$^{-1}$, which is four times higher than that in the untreated dust.' to 'The concentration of dissolved inorganic nitrogen (DIN, i.e. $NO_3^-+NO_2^-+NH_4^+$) in the AM-dust was 577 μmol g$^{-1}$ (Table 3) and increased by a factor of four against that in the untreated dust';

Line 213-214: Changed 'The N:P ratio in the AM-dust was ~134, far greater than 16 (Redfield Ratio), and similar to those reported for Asian dust aerosols in previous studies' to 'The N:P ratio of ~134 in the AM-dust was far greater than 16 (Redfield Ratio), and similar to those in Asian dust aerosols previously reported'.

Line 275: Changed 'sp.' to 'spp.'.

**Referee #1**

*General comments: This study describes the results of several experiments in which surface plankton communities from the Yellow Sea and the NW Pacific ocean were amended with atmospheric dust and different nutrients added alone and in various combinations. The responses studied included the numerical abundance of diatoms and dinoflagellates, size-fractionated chl a concentration, and nutrient concentration. The main strength of the study is that parallel incubations, in which inorganic nutrients were added in different combinations, allowed the authors to gain insight into the causative mechanisms underlying the phytoplankton responses to dust. However, in some cases (as in the case of P availability, see below) the authors seem to over-interpret the available evidence. A limitation of the study is that only standing stocks were examined; no metabolic rate measurements were included, and therefore it is not possible to ascertain the dominant type of nutrient limitation (Blackman versus Liebig). The authors rely heavily on the use of chl a as a proxy for phytoplankton biomass. However, variability in C:Chla ratios should be taken into account. The extent to which the simulated process of atmospherical tranformation of dust yields materials that are realistic in terms of nutrient content and solubility should be discussed. Finally, some sections of the Discussion are speculative and based on tenuous assumptions. All these limitations should be addressed before publication is recommended in Biogeosciences. Some suggestions as to data presentation and analysis are also given below.*

Response:

We would like to thank the referee very much for the valuable comments which enabled us to improve the quality of the manuscript. We have revised the manuscript accordingly to address the comments.

Regarding concern over nutrient limitation status: The standing stocks such as Chl *a* and biomass have been widely used for ascertaining the dominant type of nutrient limitation in the ocean. However, indeed, we acknowledge that metabolic rate has a better representation than standing stocks in ascertaining nutrient limitation. We will add metabolic rate measurements for future incubation experiments.

Regarding C;Chl *a* ratios, we have illustrated that our results are based on Chl *a* and cell abundance rather than biomass, and also considered the uncertainty in the new manuscript (refer to Q4).

We have provided additional descriptions about the simulating processes in Text S1 (refer to Q1).

Regarding concerns over P, we have made a detailed response to Q5.

*Specific comments*

*1. Simulation of atmospheric transformation of dust. How do these AM-dust materials compare, in terms of nutrient composition and solubility, with real dust samples collected in situ? This is critical to assess if the results observed are representative of real responses at sea. Table in supp. info. shows that N concentration is increased 4 orders of magnitude relative to N concentration in collected rain. Does this mean that the potential for nutrient supply is grossly overestimated in these artificially treated materials?*

Response:

It is well known that the dust deposition can supply bioavailable nutrients such as N, P, and Fe to support the growth of phytoplankton. Therefore, we mainly focused on the effect of N, P, and Fe supplied by AM-dust on phytoplankton growth in this study. During a strong Asian dust event, the loading of inorganic nitrogen ($NO_3^-+NH_4^+$), soluble P and Fe in atmospheric particles collected in the Yellow Sea was ~714 μmol $g^{-1}$, ~4.3 μmol $g^{-1}$ and ~550 μg $g^{-1}$, respectively (Shi et al., 2012). The values in AM-dust used in this study are 577 μmol $g^{-1}$, 4.3 μmol $g^{-1}$ and 473 μg $g^{-1}$, respectively (Table 3 in the origin version), which were highly comparable to those observed by Shi et al. (2012). The authors, however, agree that the loadings of these components in atmospheric dust particles could highly vary in different cases. For instance, the content of DIN and soluble P in Asian dust aerosols after a long-range transport varied over a range of two-three orders of magnitude, e.g., 11-3253 μmol $g^{-1}$ for DIN and 0.26-18.86 μmol $g^{-1}$ for soluble P (Liu et al., 2013; Meng, et al., 2016; Qi, et al., 2017). Besides, trace metals Fe, Mn, and Co were mainly originated from mineral aerosols, thus we can observe noticeably enhanced solubility of Fe, Mn, and Co in the AM-dust relative to untreated dust. The contents of other soluble trace metals such as Cu, Pb, and Zn in the dust were mainly affected by anthropogenic factor such as automobile exhaust and coal combustion, which were not reflected well in our study. To this point, we have added an illustration at the end of the conclusion. (Line 497-500)

We have added a detailed description of preparing AM-dust in the Text S1 to illustrate the difference in N content in the AM-dust and the collected rain (Line 755-765):

In this study, the aging process of dust followed Guieu's (2010) method and aimed at stimulating the cloud reaction between dust and synthetic evaporating cloud water. The pH around dust in the cloud process (i.e. mix with evaporating cloud water) was found to be as low as ~1 during their transport to the Yellow Sea (Meskhidze et al., 2003), whereas the typical pH in rainwater is 5 (Watanabe et al. 2001, Sasakawa and Uematsu, 2002, Wang et al. 2002, Sakihama et al. 2008, Zhang et al. 2011), meaning that a dilution by a factor of $10e^4$. In consequent, in order to reproduce an evaporating cloud, we have used a concentration that is 10 000-fold larger in our experiments than the typical concentrations found in rainwater. Considering the typical concentrations of dust in rainwaters was 10 mg $L^{-1}$ (Ridame et al., 2002), the dust loading in evaporating cloud water could reach 100 g $L^{-1}$. As a consequence, all of the concentrations in evaporating cloud water were around 10000-fold larger (i.e. 4 orders of magnitude larger) than those in natural rainwater. Table S1 summarized the primary chemical composition of rains in the Eastern Asian regions and the evaporating cloud water used for our simulation.

Shi, J., Gao, H., Zhang, J., Tan, S., Ren, J., Liu, C., Liu, Y., and Yao, X.: Examination of causative link between a

spring bloom and dry/wet deposition of Asian dust in the Yellow Sea, China, Journal of Geophysical Research: Atmospheres, 117, doi:10.1029/2012JD017983, 2012.

Liu, Y., Zhang, T., Shi, J., Gao, H., and Yao, X.: Responses of chlorophyll a to added nutrients, Asian dust, and rainwater in an oligotrophic zone of the Yellow Sea: Implications for promotion and inhibition effects in an incubation experiment, Journal of Geophysical Research: Biogeosciences, 118, 1763-1772, doi: 10.1002/2013JG002329, 2013.

Meng, X., Chen, Y., Wang, B., Ma, Q., and Wang, F.: Responses of phytoplankton community to the input of different aerosols in the East China Sea, Geophysical Research Letters, 43, 7081-7088, doi: 10.1002/2016GL069068, 2016.

Qi, J., Zhang, R., Chen, X., Lin, X., Gao, H., and Liu, R.: The concentration, source apportionment and deposition flux of atmospheric particulate inorganic nitrogen during dust events. Atmospheric Chemistry & Physics. Discussions, doi: 10.5194/acp-2016-1183, 2017.

Guieu, C., Dulac, F., Desboeufs, K., Wagener, T., Pulido-Villena, E., Grisoni, J.-M., Louis, F., Ridame, C., Blain, S., and Brunet, C.: Large clean mesocosms and simulated dust deposition: a new methodology to investigate responses of marine oligotrophic ecosystems to atmospheric inputs, Biogeosciences, 7, 2765-2784, doi: 10.5194/bg-7-2765-2010, 2010.

Meskhidze, N., Chameides, W. L., Nenes, A., and Chen, G.: Iron mobilization in mineral dust: Can anthropogenic $SO_2$ emissions affect ocean productivity?. Geophysical Research Letters, 30(21), doi: 10.1029/2003GL018035, 2003.

Watanabe, K., Ishizaka, Y., and Takenaka, C.: Chemical characteristics of cloud water over the Japan Sea and the Northwestern Pacific Ocean near the central part of Japan: airborne measurements. Atmospheric Environment, 35(4), 645-655, doi: 10.1016/S1352-2310(00)00358-7, 2001.

Sasakawa, M., and Uematsu, M.: Chemical composition of aerosol, sea fog, and rainwater in the marine boundary layer of the northwestern North Pacific and its marginal seas. Journal of Geophysical Research: Atmospheres, 107(D24), doi: 10.1029/2001JD001004, 2002.

Wang, Z., Akimoto, H., and Uno, I.: Neutralization of soil aerosol and its impact on the distribution of acid rain over east Asia: Observations and model results. Journal of Geophysical Research: Atmospheres, 107(D19), doi: 10.1029/2001JD001040, 2002.

Sakihama, H., Ishiki, M., and Tokuyama, A.: Chemical characteristics of precipitation in Okinawa Island, Japan. Atmospheric Environment, 42(10), 2320-2335, doi:    10.1016/j.atmosenv.2007.12.026, 2008.

Zhang, J., Zhang, G. S., Bi, Y. F., and Liu, S. M.: Nitrogen species in rainwater and aerosols of the Yellow and East China seas: Effects of the East Asian monsoon and anthropogenic emissions and relevance for the NW Pacific Ocean. Global Biogeochemical Cycles, 25(3), doi: 10.1029/2010GB003896, 2011.

Ridame, C., and Guieu, C.: Saharan input of phosphate to the oligotrophic water of the open western Mediterranean Sea. Limnology and Oceanography, 47(3), 856-869, doi: 10.4319/lo.2002.47.3.0856, 2002.

*2. Section 3.2. This section should present first the changes in nutrient concentration, and then those of chla*

*concentration. In both cases, the actual increases (absolute values) should be described (e.g. nutrient or chl a concentration increased by xx umol/L or ug/L), rather than just the relative increases (xx-fold). It is important to describe the chla responses in terms of absolute value of increase, so that they can be compared with the amount of nutrient released from the dust or provided by the nutrient amendments.*

Response:

Agree. We have revised accordingly through the Section 3.2. (Line 219-249)

*3. Conversion efficiency index. This index should be described in the Methods section.*

Response:

Agree. It has been moved to the Materials and Methods Section 2.6. (Line 170-181)

*4. It is unclear why the index is formulated in this way. Why not use just final minus initial chla concentration, as is done for N? It does not make sense to sum consecutive differences over time in chla concentration between treatments and control. In addition, the index has a potential flaw, because C:Chla ratios are bound to be different in the different sites (due, for instance, to differences in nutrient and/or light availability). So the same response in terms of % increase in biomass (carbon) will yield higher chla concentration (in absolute values), and thus higher conversion efficiency, in waters with low phytoplankton C:Chla values. The limitations of using Chla as a proxy for biomass should be acknowledged and discussed. Finally, when reporting the values of this index in the text, its units should be indicated.*

Response:

In the revision, we have clarified the net conversion efficiency index (NCEI) proposed in this study to be an approximate estimation for the utilization of N for the growth of phytoplankton. Therefore, the capacity to synthesize Chl *a* per unit concentration of nitrogen (N) in different treatments can be compared. We agree that the sum consecutive differences over time in Chl *a* concentration between treatments and control will lead to an overestimation of the real net conversion efficiency because of the accumulation effect. Theoretically, the use of the maximum difference will lead to an underestimation of the real net conversion efficiency because of degradation of Chl *a* in the growth period. The real net conversion efficiency should be between those calculated by the two approaches. This has been clarified in the revision. (Line 179-181)

We agree that C:Chl *a* ratios can be different at the different sites and Chl *a* regarding as a proxy for biomass is not appropriate. Thus, we have illustrated that our results are based on Chl *a* and cell abundance rather than biomass, and also added a discussion illustrating the varying C:Chl *a* ratios at different sites in the revision. (Line 373-376)

The unit of NCEI in the text have been added in the revision. (Line 340-363)

*5. Section 4.3. This section is speculative and difficult to follow. It is unclear how the 'increase in bioavailable P concentration following AM-dust addition' has been identified. The relationship between N:P ratios in supply vs*

*demand is tentative at best, since actual supply N:P ratios were not measured. The paragraph on lines 395-406 starts with an untenable assumption, namely that 'C_N:P in AM-dust treatments is equal to that in N treatments'. To the extent that dust additions and N additions create distinct nutrient environments, it is most unlikely that consumption N:P ratios will be the same. In fact, the previous paragraph has argued that consumption N:P ratio is lower in dust treatments than in N treatments. Thus the subsequent calculations and conclusions have no use. This sub-section (l. 395-406) should be deleted. In the subsequent paragraph, the basis for the need for additional P supply is unclear.*

Response:

We have determined the soluble nutrients leached from AM-dust in the seawater. The details can be seen in Methods Section 2.4 (Line 139-146). The content of bioavailable nutrients from AM-dust has been listed in Table 3. As the added concentration of AM-dust was 2 mg L$^{-1}$, thus, we can get the theoretical amounts of N and P nutrients supplied by AM-dust in the incubation system. The time interval of adding materials to the incubation bottles and sampling seawater for nutrient measurement was 1-2 hr. Microbial uptake, scavenging by cell surface and bottle wall, etc., possibly decreased the concentrations of nutrients at the 1-2 hr, leading to the measured values smaller than the theoretical values, which has been proved by the previous incubation experiments (Liu et al., 2013). When the concentrations of $NO_3^-$+$NO_2^-$ and $PO_4^{3-}$ decreased in the seawater, those absorbed by cell surface and bottle wall had the potential to be released into the solution again for reaching equilibrium. Therefore, the calculated theoretical amounts of N and P nutrients supplied by AM-dust in the incubation system is close to the actual values relative to those measured directly on day 0.

We have corrected the inappropriate expression of '$C_{N:P}$ in AM-dust treatments is equal to that in N treatments' and changed the equation (2) into inequation (2). (Line 441)

The last paragraph in the Discussion Section 4.3 is particularly important because it provides a new insight to analyze the budget of bioavailable P in the AM-dust addition incubation experiments. Unfortunately, the original presentation seemed to be unclear and didn't service the target well. We have rewritten this paragraph to make our thoughts understandable. (Line 453-466)

Liu, Y., Zhang, T., Shi, J., Gao, H., and Yao, X.: Responses of chlorophyll a to added nutrients, Asian dust, and rainwater in an oligotrophic zone of the Yellow Sea: Implications for promotion and inhibition effects in an incubation experiment, Journal of Geophysical Research: Biogeosciences, 118, 1763-1772, doi: 10.1002/2013JG002329, 2013.

Minor comments

*1. lines 53-54: Rewrite sentence: 'The N:P ratio of dust deposition is much higher than the Redfield ratio*

*(N:P=16)...'*

Response:

Corrected. (Line 54)

*2. line 110. Were attenuation filters used? PAR levels should be given.*

Response:

We did not use attenuation filters in this study. The surface seawater (2-5 m) was collected and incubated under natural light (Line 110). In this study, we primarily made a comparison between groups at each station rather than that between stations. Since the experiments at the one station experienced the same conditions, the effect of different PAR levels on our comparison for the results could be reduced as much as possible. We thank the comment and will add the attenuation filters for future incubation experiments.

*3. line 139. The ultrasonic method should be described briefly. The use of ultrasounds maximises the extraction of nutrients but it probably overestimates the amount of nutrients that is actually released in real conditions at sea.*

Response:

We have made a brief description of the ultrasonic method in the revision. (Line 139-141)

In general, more than 30 minute of ultra-sonication treatment will increase the temperature and may enhance the leaching of nutrients and thus overestimate. In our study, the time duration used for ultra-sonication is 30 minute. We used the ice pack to keep the temperature of water bath stable at ~0℃. Please also refer to 'Specific comments No. 3' for Referee #3.

*4. line 141. Re-write sentence, '...and filtrates were stored...'*

Response:

Corrected. (Line 142)

*5. line 153: '...enumeration of...*

Response:

Corrected. (Line 153)

*6. line 171: delete 'evidently'.*

Response:

Corrected. (Line 197)

*7. line 174: 'trophic level' means something else. Replace by appropriate phrase.*

Response:

We have replaced 'trophic level' with 'trophic state'. (Line 200)

*8. line 185: Why is P gained during the treatment?*

Response:

The acid process can enhance the solubility of P in the mineral dust.

*9. line 189 Here cite Fig. 3, otherwise the reader does not know where is that increase reported. Tables do not report the nutrient increases observed in the treatments.*

Response:

Corrected. (Line 215)

*10. line 194 and elsewhere (including Fig. legends). The phrase 'successive increase' should be omitted. Sentence should read simply: 'During the incubation...'*

Response:

The duration of the incubation experiment in this study is 9~10 days, but we mainly analysed the data on the initial 3-5 days, in which the Chl *a* concentration showed a successive increase. Thus, we used the 'successive increase' to distinguish the initial 3-5 days from the whole incubation period (9-10 days). Please refer to the Methods Section 2.6. (Line 164-168)

*11. line 271. Remove 'certain amount of'.*

Response:

Corrected. (Line 305)

*12. line 300. This sentence seems to assume that all N present in the dust becomes bioavailable, because the concentrations referred to are those given in Table 3 (which correspond to concentrations in the dust, not in seawater).*

Response:

We have rewritten the sentence. The contents of N and P in the untreated and AM-dust listed in Table 3 were determined in seawater. The details can be seen in the description of 'ultrasonic bath'. Please also refer to 'Minor comments No. 15' for Referee #1. (Line 328)

*13. line 448-449. Here the authors are deriving biogeochemical conclusions on the functioning of the biological pump, but their data consider just phytoplankton. Without information on how the metabolic activity of heterotrophs, bacteria in particular, changes in response to nutrient/dust additions, the ultimate effect on the biological pump remains unknown.*

Response:

Thank you for your thoughtful suggestion. We have rewritten the sentence and illustrated that the enhanced biological pump was only a possibility. (Line 495-497)

*14. Table 1. Silicate measurements are missing – they would have been helpful to constrain the stoichiometry of diatom blooms in response to nutrient/dust amendments.*

Response:

In the original experimental settings, we aimed at exploring the relationship between phytoplankton growth and supplied bioavailable nutrients (N, P, and Fe) by AM-dust additions. Thus, we did not determine silicate (Si) concentrations. For the Yellow Sea, Si is generally not a limiting nutrient for the growth of phytoplankton in spring because of the influence of riverine input (Wang et al., 2003). Si may become a limiting nutrient during the diatom blooms in the open ocean of the northwest Pacific, but it will not influence the result of this study, which was mainly concerned with N, P, and Fe nutrients. We thank the comment and will add the measurement of silicate for future incubation experiments.

Wang, B. D., Wang, X. L., and Zhan, R. Nutrient conditions in the Yellow Sea and the East China Sea. Estuarine, Coastal and Shelf Science, 58, 127-136, doi: 10.1016/S0272-7714(03)00067-2, 2003.

*15. Table 3. The data labelled 'increased concentrations' are theoretical or expected concentrations, assuming 100% of the nutrients in the dust becomes dissolved. This should be explicitly acknowledged in the Table legend. Is there any evidence to support the tenet that, upon dust deposition onto the ocean's surface, all nutrients become dissolved and bioavailable?*

Response:

We have added 'theoretically' in the Table legend. (Line 696)

The $NO_3^-+NO_2^-$ in the AM-dust would dissolve almost thoroughly once exposing to the filtered seawater (Ridame et al., 2014) while the dissolution of $PO_4^{3-}$ would continue over multiple days because the dissolution of less labile (but still soluble) P compounds would take some time (generally lower than 72 hr) to dissolve in seawater (Mackey et al., 2012). Over the duration of the incubation experiments, phytoplankton in the incubated bottles can absorb the soluble $NO_3^-+NO_2^-$ and $PO_4^{3-}$ from AM-dust if they need.

Ridame, C., Dekaezemacker, J., Guieu, C., Bonnet, S., L'Helguen, S., and Malien, F.: Contrasted Saharan dust events in LNLC environments: impact on nutrient dynamics and primary production, Biogeosciences, 11, 4783-4800, doi: 10.5194/bg-11-4783-2014, 2014.
Mackey, K. R., Roberts, K., Lomas, M. W., Saito, M. A., Post, A. F., and Paytan, A.: Enhanced solubility and ecological impact of atmospheric phosphorus deposition upon extended seawater exposure, Environmental science & technology, 46, 10438-10446, doi: 10.1021/es3007996, 2012.

*16. Figure legends: The phrase 'successive increase' is awkward. Delete in all fig. legends. It should be simply: 'changes in xxxx during the incubation period at each station'.*

Response:

Please see 'Minor comments No.10'.

*17. Fig. 2. Y-axis intervals should be regular (e.g., 0.5 or 1.0 ug/L) and consistent in all plots. Minor ticks should be included, to help the reader ascertain the magnitude of responses.*

Response:

Corrected. (Line 715)

*18. Fig. 3. Symbol for control should be more visible (it is often masked by other symbols).*

Response:

Corrected. (Line 710)

*19. Fig. 5. Revise species names spelling (e.g. Skeletonema). Species names and genera should be written in italics (but not 'spp.').*

Response:

Corrected. (Line 725)

*20. Figs. 6 and 7. The index values shown here result from the subtraction and division of variables measured independently, each with its own error. Therefore the error bars shown should be computed using the error propagation formulae for addition and division.*

Response:

Thanks for the valuable suggestion. We have used the error propagation formulae to calculate the error bars and clarified in the revision. Please see Figs. 6 and 7. (Line 340-363, 739-741)

*21. Fig. 8. These N:P ratios should be defined in the Methods section. Strictly speaking, the N:P supply ratio is not known, since no solubility experiments were conducted.*

Response:

We have moved the definitions of N:P ratios to the Methods Section 2.6. The dissolved concentrations of N and P from AM-dust in the seawater have been determined in the laboratory (Table 3). Please also refer to 'Minor comments No.12 and 15'. (Line 183-186)

**Supplementary comments**

*Specific comments No. 3 for Referee #3:*

*3 Section 2.4: The ultrasonic bath treatment may overestimate the nutrient concentration. What was the time duration used for ultra-sonication? More than 30 minute of treatment will increase the temperature and may enhance the leaching of nutrients and thus overestimate. Usually, the treatment is done for aerosol samples collected on filter substrate to loosen the particles from matrix.*

Response:

The time duration used for ultra-sonication is 30 minute. We used the ice pack to keep the temperature of water bath stable at ~0°C. We have added a detailed description of ultrasonic bath treatment in the text. (Line 139-141).

[revised manuscript text omitted]

---

## Author Comment (AC3) · 6 Nov 2017

**Additional changes we made to improve the presentation:**

Line 52: Changed 'worthy' to 'worthy of';

Line 163: Changed 'Selection of study period' to 'Protocol of data analysis';

Line 200-201: Changed 'For the S5 station in the YS, although the nutrient levels were comparable to those at S1, Chl *a* concentrations were as high as 2.74 μg L$^{-1}$, close to the conditions of spring bloom in the YS' to 'Although the nutrient levels at S5 in the YS were comparable to those at S1, the Chl *a* concentration of 2.74 μg L$^{-1}$ was close to the values during spring blooms in the YS';

Line 204-205: Changed 'S1 and S2 (< 20 cells mL$^{-1}$) to 79 cells mL$^{-1}$ at S3 and 35 cells mL$^{-1}$ at S4' to '< 20 cells mL$^{-1}$ at S1 and S2 to 35 cells mL$^{-1}$ at S4 and 79 cells mL$^{-1}$ at S3';

Line 209-210: Changed 'As shown in Table 3, the concentration of dissolved inorganic nitrogen (DIN, i.e. NO$_3^-$+NO$_2^-$+NH$_4^+$) in the AM-dust was 577 μmol g$^{-1}$, which is four times higher than that in the untreated dust.' to 'The concentration of dissolved inorganic nitrogen (DIN, i.e. NO$_3^-$+NO$_2^-$+NH$_4^+$) in the AM-dust was 577 μmol g$^{-1}$ (Table 3) and increased by a factor of four against that in the untreated dust';

Line 213-214: Changed 'The N:P ratio in the AM-dust was ~134, far greater than 16 (Redfield Ratio), and similar to those reported for Asian dust aerosols in previous studies' to 'The N:P ratio of ~134 in the AM-dust was far greater than 16 (Redfield Ratio), and similar to those in Asian dust aerosols previously reported'.

Line 275: Changed 'sp.' to 'spp.'.

**Referee #2**

*General comments*

*1. Biological productivity of the open ocean regions, especially oligotrophic parts, have attracted the attention of several researchers. Our present understanding of the role of mineral dust in enhancing the primary productivity of the oligotrophic ocean by supplying bio-available nutrients is still in its infancy. The present manuscript aims at advancing such an understanding of enhancement of phytoplankton growth and the phytoplankton community structure by the nutrient supply from the Asian in the subtropical gyre, Kuroshio Extension of Northwest Pacifica Ocean and Yellow Sea region. The authors attempt this by incubation experiment onboard R/V Dongfanghong II during spring. For incubation experiment, the mineral dust collected from Gobi Desert region was artificially modified and phytoplankton assemblages from subtropical gyre, Northwest Pacific ocean and Yellow Sea region were used. Each of the five microcosm experiment lasted for 9 to 10 days. Using a net conversion efficiency index, proposed by the authors, of nitrogen conversion to chlorophyll a the authors explore the role of bio-available nutrients from mineral dust in the primary production at the above mentioned three regions.*
*2. The subject matter of the manuscript addresses an important aspect of phytoplankton growth by "bio-available nutrients" from "treated soil from Gobi desert" which is "expected to" simulate the natural mineral dust. Though there are several concerns that have been listed under specific comments, in my opinion, the results are publishable; but only after adequately addressing the concerns.*

Response:

We greatly appreciate the referee for the thoughtful comments and have revised our manuscript accordingly.

Major concerns:

*1. The major concern is that the authors have not succeeded in unambiguously resolving the issue of quantification of bioavailability of nutrients form artificially modified mineral dust, especially phosphorous, for phytoplankton growth, which is the central theme of the manuscript. See for example, lines 334-341. The ambiguity regarding the 'missing N and P'.*

Response:

We are sorry that the word of 'missing' in the manuscript confused the referee. The 'missing N and P' and 'missing parts' in the text meant the N and P might be adsorbed on the bottle walls, suspended particles, and phytoplankton in the solution, while 'missing sources' meant there might be other sources to supply bioavailable P. Thus, we will change 'missing sources' to 'other sources' to express them clearly (Line 482). Considering the comment and the

similar comments from other two referees, we have rewritten the part related to the budget of P in incubation experiments. Please see our revised manuscript. (Line 419-466)

*2. How would the authors differentiate the phytoplankton growth-response due to N/P/Fe and that due to Mn and Zn (see for example, Saito et al., 2008; Sunda 2012).*
Response:
We have replaced "Zn" with "Co" in the text because the concentration of suite of trace metals in AM-dust are very low except Fe, Mn and Co (Table 3). Please also see Specific Comments No. 4 for Referee #3. (Line 359)

We used the various nutrient addition experiments to illustrate which nutrients (i.e. N, P, and Fe) limit phytoplankton growth. As we did not conduct Mn or Co addition incubation experiments and determine the concentration of Mn and Co in the seawater, it is difficult for us to accurately see whether phytoplankton were under limitation of Mn and/or Co. However, through making a comparison of the net conversion efficiency index (NCEI) in various nutrient addition treatments, we can roughly determine whether there were other nutrients except N, P, and Fe dissolved from AM-dust to stimulate phytoplankton growth. For instance, the NCEI value in AM-dust treatment was significantly higher than that in N+P+Fe treatments, indicating the N, P, and Fe supplied by AM-dust cannot explain the corresponding phytoplankton growth. Besides, As reported in literature, trace metals such as Mn and Co can affect phytoplankton growth rate (Sunda, 2012), which might lead to higher NCEI. The effect of Mn and Co on phytoplankton growth is just a hypothesis, which have been clarified in the revision. (Line 359-360)

Sunda, W.: Feedback interactions between trace metal nutrients and phytoplankton in the ocean, Frontiers in microbiology, 3, 204, doi: 10.3389/fmicb.2012.00204, 2012.

*3. The authors do not observe any community shift in the phytoplankton in their study. There is no discussion on this aspect and authors need to address this.*
Response:
We thank the comment and have added some discussions about community shift of phytoplankton in Discussion Section 4.2, as well as relative abundance in Fig. 5. (Line 402-412, Line 725)

*4. Based on the information provided, it is hard to see how closely the artificially modified soil collected from Gobi desert mimics the nature. Some more robust information on the atmospheric (chemical) processing of the mineral dust during the long-range transport from source to the proposed study site during spring is needed along with the upper air wind vectors. What are the chemical constituents of such atmospheric processed mineral dust in presence of anthropogenic aerosol is not clear.*
Response:
It is well known that the dust deposition can supply bioavailable nutrients such as N, P, and Fe to support the growth of phytoplankton. Therefore, we mainly focused on the effect of N, P, and Fe supplied by AM-dust on phytoplankton

growth in this study. During a strong Asian dust event, the loading of inorganic nitrogen ($NO_3^-$+$NH_4^+$), soluble P and Fe in atmospheric particles collected in the Yellow Sea was ~714 μmol g$^{-1}$, ~4.3 μmol g$^{-1}$ and ~550 μg g$^{-1}$, respectively (Shi et al., 2012). The values in AM-dust used in this study are 577 μmol g$^{-1}$, 4.3 μmol g$^{-1}$ and 473 μg g$^{-1}$, respectively (Table 3 in the origin version), which were highly comparable to those observed by Shi et al. (2012). The authors, however, agree that the loadings of these components in atmospheric dust particles could highly vary in different cases. For instance, the content of DIN and soluble P in Asian dust aerosols after a long-range transport varied over a range of two-three orders of magnitude, e.g., 11-3253 μmol g$^{-1}$ for DIN and 0.26-18.86 μmol g$^{-1}$ for soluble P (Liu et al., 2013; Meng, et al., 2016; Qi, et al., 2017). Besides, trace metals Fe, Mn, and Co were mainly originated from mineral aerosols, thus we can observe noticeably enhanced solubility of Fe, Mn, and Co in the AM-dust relative to untreated dust. The contents of other soluble trace metals such as Cu, Pb, and Zn in the dust were mainly affected by anthropogenic factor such as automobile exhaust and coal combustion, which were not reflected well in our study. To this point, we have added an illustration at the end of the conclusion. (Line 497-500)

We have also added a description of preparing AM-dust in the Text S1 to illustrate atmospheric processing of the mineral dust during the long-range transport (Line 755-765):

In this study, the aging process of dust followed Guieu's (2010) method and aimed at stimulating the cloud reaction between dust and synthetic evaporating cloud water. The pH around dust in the cloud process (i.e. mix with evaporating cloud water) was found to be as low as ~1 during their transport to the Yellow Sea (Meskhidze et al., 2003), whereas the typical pH in rainwater is 5 (Watanabe et al. 2001, Sasakawa and Uematsu, 2002, Wang et al. 2002, Sakihama et al. 2008, Zhang et al. 2011), meaning that a dilution by a factor of 10e$^4$. In consequent, in order to reproduce an evaporating cloud, we have used a concentration that is 10 000-fold larger in our experiments than the typical concentrations found in rainwater. Considering the typical concentrations of dust in rainwaters was 10 mg L$^{-1}$ (Ridame et al., 2002), the dust loading in evaporating cloud water could reach 100 g L$^{-1}$. As a consequence, all of the concentrations in evaporating cloud water were around 10000-fold larger (i.e. 4 orders of magnitude larger) than those in natural rainwater. Table S1 summarized the primary chemical composition of rains in the Eastern Asian regions and the evaporating cloud water used for our simulation.

Shi, J., Gao, H., Zhang, J., Tan, S., Ren, J., Liu, C., Liu, Y., and Yao, X.: Examination of causative link between a spring bloom and dry/wet deposition of Asian dust in the Yellow Sea, China, Journal of Geophysical Research: Atmospheres, 117, doi:10.1029/2012JD017983, 2012.

Liu, Y., Zhang, T., Shi, J., Gao, H., and Yao, X.: Responses of chlorophyll a to added nutrients, Asian dust, and rainwater in an oligotrophic zone of the Yellow Sea: Implications for promotion and inhibition effects in an incubation experiment, Journal of Geophysical Research: Biogeosciences, 118, 1763-1772, doi: 10.1002/2013JG002329, 2013.

Meng, X., Chen, Y., Wang, B., Ma, Q., and Wang, F.: Responses of phytoplankton community to the input of different aerosols in the East China Sea, Geophysical Research Letters, 43, 7081-7088, doi:

10.1002/2016GL069068, 2016.

Qi, J., Zhang, R., Chen, X., Lin, X., Gao, H., and Liu, R.: The concentration, source apportionment and deposition flux of atmospheric particulate inorganic nitrogen during dust events. Atmospheric Chemistry & Physics. Discussions, doi: 10.5194/acp-2016-1183, 2017.

Guieu, C., Dulac, F., Desboeufs, K., Wagener, T., Pulido-Villena, E., Grisoni, J.-M., Louis, F., Ridame, C., Blain, S., and Brunet, C.: Large clean mesocosms and simulated dust deposition: a new methodology to investigate responses of marine oligotrophic ecosystems to atmospheric inputs, Biogeosciences, 7, 2765-2784, doi: 10.5194/bg-7-2765-2010, 2010.

Meskhidze, N., Chameides, W. L., Nenes, A., and Chen, G.: Iron mobilization in mineral dust: Can anthropogenic $SO_2$ emissions affect ocean productivity?. Geophysical Research Letters, 30(21), doi: 10.1029/2003GL018035, 2003.

Watanabe, K., Ishizaka, Y., and Takenaka, C.: Chemical characteristics of cloud water over the Japan Sea and the Northwestern Pacific Ocean near the central part of Japan: airborne measurements. Atmospheric Environment, 35(4), 645-655, doi: 10.1016/S1352-2310(00)00358-7, 2001.

Sasakawa, M., and Uematsu, M.: Chemical composition of aerosol, sea fog, and rainwater in the marine boundary layer of the northwestern North Pacific and its marginal seas. Journal of Geophysical Research: Atmospheres, 107(D24), doi: 10.1029/2001JD001004, 2002.

Wang, Z., Akimoto, H., and Uno, I.: Neutralization of soil aerosol and its impact on the distribution of acid rain over east Asia: Observations and model results. Journal of Geophysical Research: Atmospheres, 107(D19), doi: 10.1029/2001JD001040, 2002.

Sakihama, H., Ishiki, M., and Tokuyama, A.: Chemical characteristics of precipitation in Okinawa Island, Japan. Atmospheric Environment, 42(10), 2320-2335, doi: 10.1016/j.atmosenv.2007.12.026, 2008.

Zhang, J., Zhang, G. S., Bi, Y. F., and Liu, S. M.: Nitrogen species in rainwater and aerosols of the Yellow and East China seas: Effects of the East Asian monsoon and anthropogenic emissions and relevance for the NW Pacific Ocean. Global Biogeochemical Cycles, 25(3), doi: 10.1029/2010GB003896, 2011.

Ridame, C., and Guieu, C.: Saharan input of phosphate to the oligotrophic water of the open western Mediterranean Sea. Limnology and Oceanography, 47(3), 856-869, doi: 10.4319/lo.2002.47.3.0856, 2002.

*5. It is not clear from the manuscript, whether the ocean atmospheric conditions during the 3-month period (March-May 2014) at each of the sampling locations could be considered as a part of the same season where ocean and atmosphere represents similar conditions. The data from the Table 1 do not support this. For example, the average temperature (is it SST?) at S4 is quite different from the rest of the stations, which was sampled in May 2014. Similar, the MLD also is quite different. The authors need to address these issues.*

Response:

We are sorry that we made a mistake of sea surface temperature (SST) and mixed layer depth (MLD) at S5 and have corrected them in the revision. (Line 697)

As a general idea on season, those stations were sampled almost in the same season. It is the location not the time that induces a low temperature at S4 and S5 but a high temperature at the other three stations (S1-S3).

Stations S4 and S5 are located in a shallow coastal area with a depth less than 100 m. The winter cooling induces a vertically homogenous water temperature with a value around 12 °C (Yeh & Kim, 2010). The heating process usually starts in early April and there is no much time to increase the water temperature before our observations. Consequently, we observed a low temperature at the two stations in April and May respectively (the difference between S4 and S5 is also caused by location: S4 is north to S5 and therefore has a lower temperature than S5 in winter; S4 is close to the Bohai Strait and therefore has a strong vertical mixing than S5 that prevents increasing of SST after April.)

On the other hand, stations S1-S3 are located at the open water with a depth larger than 2000 m. The winter cooling cannot easily reduce its water temperature because of the heat supplying from bottom of mixed layer (the sea bed in coastal water has not such function). In addition, the *Kuroshio* and mesoscale eddies also bring warm water into the observation area. Such lateral supply of heat can also prevent the decreasing of surface temperature in winter. This is why we observed a temperature around 19 °C.

The MLD depends on wind stresses and heat fluxes over the sea. The surrounding land induces weaker winds over stations S4 and S5 than over stations S1-S3. Consequently, the mixed layer is shallower at stations S4 and S5 than at station S1-S3. In addition, the larger loss of latent and sensible heats over stations S1-S3 (due to higher SST) than over stations S4 and S5 is also likely responsible for the deeper mixed layer at stations S1-S3 than at station S4 and S5.

In this study, we primarily made a comparison between groups at each station rather than that between stations. Since the experiments at the one station experienced the same conditions, the different oceanic and atmospheric conditions between stations would not influence our comparison for the results.

We have added a description of SST and MLD conditions at the sampling locations in Results Section 3.1. (Line 190-195)

Yeh S W and Kim C H: Recent warming in the Yellow/East China Sea during winter and the associated atmospheric circulation, Continental Shelf Research, 30(13): 1428-1434, doi: 10.1016/j.csr.2010.05.002, 2010.

*7. Most of the results obtained from the incubation experiment, such as co-limitation of nutrients, response of phytoplankton biomass and structure, are largely known as could be seen from the literature cited in the manuscript. For example, co-limitation in the south China Sea and its response to aeolian input (Gao et al., 2012), Fu et al. (2009) on N:P ratios during spring, Fu et al (2009) study on the phytoplankton biomass and structure in South*

*China Sea. Nishibe et al. (2015) work in the Kuroshio Extension during spring. See also under Minor comments (8).*

Response:

We recognize that previous studies have identified the nutrient limitation status in the study area. We are content that our nutrient addition results are consistent with theirs, as this confirms that the conditions we were studying were representative to the regions.

The similar results compared to other studies aimed at illustrating the following ideas:

The main strength of the study is to gain insight into the causative mechanisms underlying the phytoplankton responses to dust through parallel incubations, in which inorganic nutrients were added in different combinations. The added amount of inorganic nutrients (e.g. N: 2 μmol $L^{-1}$, P: 0.2 μmol $L^{-1}$, Fe: 2 nmol $L^{-1}$) was not equal to that dissolved from AM-dust, which leads to incomparable increases of Chl *a* concentrations in AM-dust and various nutrient treatments. Thus, we proposed the conversion efficiency index (NCEI) to quantify the role of N, P, and Fe dissolved from AM-dust played in stimulating phytoplankton growth, based on making a comparison with the results concluded from various nutrient addition treatments. Finally, we used the correlation of $S_{N:P}$ and $C_{N:P}$ to highlight increased bioavailability of P in AM-dust addition experiments.

Minor concerns:

*8. Lines 310-311: Authors need to at least briefly state what are those "Complex hydrographic conditions".*

Response:

We have added a brief state of complex hydrographic conditions in the revision: "Complex hydrographic conditions e.g., various riverine inputs and atmospheric deposition, create a large spatiotemporal variation in nutrient concentrations in the YS". (Line 313)

*9. Lines 339-342: This is purely speculative and needs further substantiation.*

Response:

We have rewritten this sentence to express it clearly. (Line 328-337)

In the field incubation experiments, the time interval of adding materials (e.g. AM-dust and various nutrients) into the incubation bottles and seawater sampling for nutrient measurement (i.e. measurement on day 0) was around 1-2 hr. During this period, microbial uptake, scavenging by cell surface and bottle wall are all possible to influence the measurement of added N and P nutrients from AM-dust, which lead to the concentrations determined on day 0 were lower than the theoretical adding amounts. When the concentrations of $NO_3^-+NO_2^-$ and $PO_4^{3-}$ decreased in the seawater, those absorbed by cell surface and bottle wall had the potential to be released into the solution again for

reaching equilibrium.

*10. Lines 307-308: So what is new/different from the work of Nishibe et al. (2015).*
Response:
Please see Major concerns No.7.

*11. Line 143: Expand SPSS*
Response:
Corrected. (Line 129)

*12. Table 3 : 2ns foot note (b) is missing in the Table*
Response:
Corrected. (Line 700)

*13. Also explain "E-3, E-4"*
Response:
We have added an explanation of 'E-3' and 'E-4' in Table 3. (Line 700)

**Supplementary comments**

*Specific comments No. 4 for Referee #3::*

*4. Section 4.1: Line 328-329: This is a speculative statement and of course, it need further investigation. Except here, the role of trace metals (as nutrient or toxicant) is nowhere discussed. The concentration of suite of trace metals (in AM-dust) are very low except Fe and Mn. This once again indicate that, AM-dust is not the best representative for processed dust.*
Response:
The sentence has been revised as "Apart from N, P, and Fe, AM-dust also provided considerable other nutrients, e.g., Mn and Co (Table 3), which may have contributed to the phytoplankton growth in the incubations at S2 (Coale, 1991; Jakuba et al., 2008; Saito et al., 2008; Sunda, 2012)." The concentration of Co in the surface seawater of the Pacific generally maintained at picomolar levels (Sunda, 2012; Biller and Bruland, 2012), and the supplied Co by AM-dust in this study reached ~90 pmol $L^{-1}$, which cannot be negligible compared to the baseline values. (Line 359-361)

We have also revised the part related to the role of trace metals (as nutrient or toxicant) other than Fe, Mn and Co. As showed in our Table 3, most of trace metals are indeed negligible. (Line 458)

The aging process of dust in this study focused on the reaction between dust and inorganic acids ($H_2SO_4$ and $HNO_3$, details can be seen in Text S1). The distinguishing characteristic of AM-dust relative to untreated dust mainly reflects in the increased contents of soluble N, P, and trace metals. Trace metals Fe, Mn, and Co were mainly originated from mineral aerosols, thus we can observe noticeably enhanced solubility of Fe, Mn, and Co in the AM-dust relative to untreated dust. The contents of other soluble trace metals such as Cu, Pb, and Zn in the dust were mainly affected by anthropogenic factor such as automobile exhaust and coal combustion, which were not reflected well in our study. To this point, we have added an illustration at the end of the conclusion. (Line 497-500)

[revised manuscript text omitted]
, the aging process of dust followed Guieu's (2010) method and aimed at stimulating the cloud reaction between dust and synthetic evaporating cloud water. The pH around dust in the cloud process (i.e. mix with evaporating cloud water) was found to be as low as ~1 during their transport to the Yellow Sea (Meskhidze et al., 2003), whereas the typical pH in rainwater is 5 (Watanabe et al. 2001, Sasakawa and Uematsu, 2002, Wang et al. 2002, Sakihama et al. 2008, Zhang et al. 2011), meaning that a dilution by a factor of $10e^4$. In consequent, in order to reproduce an evaporating cloud, we have used a concentration that is 10 000-fold larger in our experiments than the typical concentrations found in rainwater. Considering the typical concentrations of dust in rainwaters was 10 mg $L^{-1}$ (Ridame et al., 2002), the dust loading in evaporating cloud water could reach 100 g $L^{-1}$. As a consequence, all of the concentrations in evaporating cloud water were around 10000-fold larger (i.e. 4 orders of magnitude larger) than those in natural rainwater. Table S1 summarized the primary chemical composition of rains in the Eastern Asian regions and the evaporating cloud water used for our simulation. As the uptake of organic acidic gases during transport is complicated for Asian dust, we did not add oxalic acid, which was used for simulating the Saharan dust by Guieu et al. (2010), to simplify the reaction of dust surface and emphasize the importance of inorganic acids ($H_2SO_4$ and $HNO_3$) (Fan et al., 2006; Formenti et al., 2011; Shi et al., 2012).

**Table S1.** Primary chemical composition of the rains in the eastern Asian region and the simulated eastern Asian cloud water.

| | pH | $NO_3^-$ (M) | $SO_4^{2-}$ (M) |
|---|---|---|---|
| Reference eastern Asian rains[*] | 3.89–7.61 | $10^{-5}$ | $10^{-5}$ |
| Simulated cloud water | 1[**] | $10^{-1}$ | $10^{-1}$ |

810    [*]Sasakawa and Uematsu, 2002; Watanabe et al. 2001; Zhang et al. 2011; Sakihama et al. 2008; Wang et al. 2002.

[**] Meskhidze et al., 2003.

**Table S2**. Recovery yield, accuracy, and detection limit for trace metal analysis

| Metal | Detection limit ($\mu$g L$^{-1}$)[*] | Recovery (%) | RSD (%)[**] |
|-------|----------------------|--------------|---------|
| Zn | 0.012 | 90.6 | 3.17 |
| Cu | 0.226 | 95.2 | 2.09 |
| Cd | 0.016 | 88.5 | 0.87 |
| Pb | 0.019 | 93.2 | 2.93 |
| Co | 0.017 | 97.9 | 0.24 |
| Fe | 3.738 | 95.4 | 3.88 |
| Mn | 0.056 | 90.9 | 4.48 |

[*] Detection limit was calculated as three times the standard deviation of the blank.

815                        [**] RSD means 'Relative Standard Deviation'.

[Figure]

**Figure S1.** Changes in Chl *a* during the incubation experiments at each station. The successive increase during the incubation period in this study is identified by the dotted line.

820

[Figure]

**Figure S2.** The relationship between the consumed N:P ratio ($C_{N:P}$) and supply N:P ratio ($S_{N:P}$) in the control and the various nutrient treatments during the successive increase in the incubation period at each station

---

## Author Comment (AC4) · 6 Nov 2017

**Additional changes we made to improve the presentation:**

Line 52: Changed 'worthy' to 'worthy of';

Line 163: Changed 'Selection of study period' to 'Protocol of data analysis';

Line 200-201: Changed 'For the S5 station in the YS, although the nutrient levels were comparable to those at S1, Chl *a* concentrations were as high as 2.74 μg L$^{-1}$, close to the conditions of spring bloom in the YS' to 'Although the nutrient levels at S5 in the YS were comparable to those at S1, the Chl *a* concentration of 2.74 μg L$^{-1}$ was close to the values during spring blooms in the YS';

Line 204-205: Changed 'S1 and S2 (< 20 cells mL$^{-1}$) to 79 cells mL$^{-1}$ at S3 and 35 cells mL$^{-1}$ at S4' to '< 20 cells mL$^{-1}$ at S1 and S2 to 35 cells mL$^{-1}$ at S4 and 79 cells mL$^{-1}$ at S3';

Line 209-210: Changed 'As shown in Table 3, the concentration of dissolved inorganic nitrogen (DIN, i.e. NO$_3^-$+NO$_2^-$+NH$_4^+$) in the AM-dust was 577 μmol g$^{-1}$, which is four times higher than that in the untreated dust.' to 'The concentration of dissolved inorganic nitrogen (DIN, i.e. NO$_3^-$+NO$_2^-$+NH$_4^+$) in the AM-dust was 577 μmol g$^{-1}$ (Table 3) and increased by a factor of four against that in the untreated dust';

Line 213-214: Changed 'The N:P ratio in the AM-dust was ~134, far greater than 16 (Redfield Ratio), and similar to those reported for Asian dust aerosols in previous studies' to 'The N:P ratio of ~134 in the AM-dust was far greater than 16 (Redfield Ratio), and similar to those in Asian dust aerosols previously reported'.

Line 275: Changed 'sp.' to 'spp.'.

**Referee #3**

*General comments*

*This paper present a set of microcosm experiments performed on-board using sea water collected from two distinct oceanic regime: 1) Oligotrophic waters (Northwest Pacific Ocean) and 2) nutrient rich waters (Yellow Sea), to understand the impact of atmospheric dust (or processed dust) and other nutrients (N, P, Fe e.t.c) on the phytoplankton productivity in terms of increase in chlorophyll a (Chl a) and abundance of phytoplankton in various size fractions (e.g. micro, nano, pico e.t.c). The artificially modified atmospheric dust (AM-dust) is prepared using surface soil collected from Gobi Desert. The set of experiments lasted for 9-10 days and clearly indicate an overall increase in Chl a concentrations due to addition of various combination of nutrients including AM-dust. Authors have proposed a "new" net conversion efficiency index (NCEI) to better understand the impact of specific nutrients (N, P, Fe, and AM-dust) on primary productivity at the sampled locations. The presented case study makes an important contribution towards improving our understanding on impact of aeolian deposition (external source of nutrients) on productivity and ocean biogeochemistry. The results obtained from set of experiments are discussed well, manuscript is easy to read (except few sections, see in specific comments) and should be of great interest to the Biogeoscience community. So, I recommend this paper for publication in Biogeosciences, but after addressing some of the concerns detailed in specific comments.*

Response:

We greatly appreciate the referee for the constructive comments and have revised our manuscript accordingly.

*Specific comments:*

*1 Section 2.1: What is the size distribution of soil samples used for preparing AM-dust? This information is important because if, majority of collected soil particle are in coarse fraction (e.g. more than 30 microns), most of them gets deposited at the source region and hardly get transported to the Pacific. So, the soil used for AMdust preparation is not at all a representative undergoing long-range transport and depositing on surface waters. The fine fraction (less than 5 microns) or typically clay fraction of the soil is a more representative dust which can be artificially processed to mimic the processed aeolian dust.*

Response:

Recently, the transport routes of Asian dust move the northward. We had practical difficulty to collect the sufficient amount of ambient dust samples for incubation experiments. Alternatively, we used AM-dust for experiments as

those reported by Guieu and Ridame, etc (Guieu et al., 2010; Ridame et al., 2014). In this study, we did not determine the size distribution of soil samples, but only the fraction less than 20 μm was used for preparing AM-dust in this study. Fe and P composition in $PM_{10}$ and $PM_{20}$ generated from the same soils were reported to be quite similar (Shi et al., 2011; Nenes et al., 2011). We agree that finer dust, e.g., less than 5 microns, should be more representative of those in the aeolian dust transported to the sea. However, it is also practically difficult to gain the sufficient amount of the fine dust for modification. Thus, it is a practical compromise by using artificially modified $PM_{20}$.

Guieu, C., Dulac, F., Desboeufs, K., Wagener, T., Pulido-Villena, E., Grisoni, J.-M., Louis, F., Ridame, C., Blain, S., and Brunet, C.: Large clean mesocosms and simulated dust deposition: a new methodology to investigate responses of marine oligotrophic ecosystems to atmospheric inputs, Biogeosciences, 7, 2765-2784, doi: 10.5194/bg-7-2765-2010, 2010.

Ridame, C., Dekaezemacker, J., Guieu, C., Bonnet, S., L'Helguen, S., and Malien, F.: Contrasted Saharan dust events in LNLC environments: impact on nutrient dynamics and primary production, Biogeosciences, 11, 4783-4800, doi: 10.5194/bg-11-4783-2014, 2014.

Nenes, A., Krom, M. D., Mihalopoulos, N., Cappellen, P., Shi, Z., Bougiatioti, A., Zarmpas, P., and Herut, B.: Atmospheric acidification of mineral aerosols: a source of bioavailable phosphorus for the oceans, Atmospheric Chemistry and Physics, 11(13): 6265-6272, doi: 10.5194/acp-11-6265-2011, 2011.

Shi, Z., Bonneville, S., Krom, M. D., Carslaw, K.S., Jickells, T. D., Baker, A.R., and Benning L.G.: Iron dissolution kinetics of mineral dust at low pH during simulated atmospheric processing, Atmospheric Chemistry and Physics, 11(3): 995-1007, doi: 10.5194/acp-11-995-2011, 2011.

*2. Section 2.2: Line 110-112: Why Day 1 was not sampled?*
Response:
In our previous incubation experiments, there was no distinct difference of Chl *a* concentrations on day 1 and day 0 in all cases (Liu et al., 2013). Thus, we did not take a sample on day 1 in this study. We agree that the data on day 1 may be valuable and will change back our sampling protocol for future incubation experiments.

Liu, Y., Zhang, T., Shi, J., Gao, H., and Yao, X.: Responses of chlorophyll a to added nutrients, Asian dust, and rainwater in an oligotrophic zone of the Yellow Sea: Implications for promotion and inhibition effects in an incubation experiment, Journal of Geophysical Research: Biogeosciences, 118, 1763-1772, doi: 10.1002/2013JG002329, 2013.

*3 Section 2.4: The ultrasonic bath treatment may overestimate the nutrient concentration. What was the time duration used for ultra-sonication? More than 30 minute of treatment will increase the temperature and may enhance the leaching of nutrients and thus overestimate. Usually, the treatment is done for aerosol samples collected on filter substrate to loosen the particles from matrix.*
Response:

The time duration used for ultra-sonication is 30 minute. We used the ice pack to keep the temperature of water bath stable at ~0°C. We have added a detailed description of ultrasonic bath treatment in the text. (Line 139-141)

*4. Section 4.1: Line 328-329: This is a speculative statement and of course, it need further investigation. Except here, the role of trace metals (as nutrient or toxicant) is nowhere discussed. The concentration of suite of trace metals (in AM-dust) are very low except Fe and Mn. This once again indicate that, AM-dust is not the best representative for processed dust.*

Response:

The sentence has been revised as "Apart from N, P, and Fe, AM-dust also provided considerable other nutrients, e.g., Mn and Co (Table 3), which may have contributed to the phytoplankton growth in the incubations at S2 (Coale, 1991; Jakuba et al., 2008; Saito et al., 2008; Sunda, 2012)." The concentration of Co in the surface seawater of the Pacific generally maintained at picomolar levels (Sunda, 2012; Biller and Bruland, 2012), and the supplied Co by AM-dust in this study reached ~90 pmol $L^{-1}$, which cannot be negligible compared to the baseline values. (Line 359-361)

We have also revised the part related to the role of trace metals (as nutrient or toxicant) other than Fe, Mn and Co. As showed in our Table 3, most of trace metals are indeed negligible. (Line 458)

The aging process of dust in this study focused on the reaction between dust and inorganic acids ($H_2SO_4$ and $HNO_3$, details can be seen in Text S1). The distinguishing characteristic of AM-dust relative to untreated dust mainly reflects in the increased contents of soluble N, P, and trace metals. Trace metals Fe, Mn, and Co were mainly originated from mineral aerosols, thus we can observe noticeably enhanced solubility of Fe, Mn, and Co in the AM-dust relative to untreated dust. The contents of other soluble trace metals such as Cu, Pb, and Zn in the dust were mainly affected by anthropogenic factor such as automobile exhaust and coal combustion, which were not reflected well in our study. To this point, we have added an illustration at the end of the conclusion. (Line 497-500)

Sunda, W.: Feedback interactions between trace metal nutrients and phytoplankton in the ocean, Frontiers in microbiology, 3, 204, doi: 10.3389/fmicb.2012.00204, 2012.
Biller, D. V., and Kenneth W. B.: Analysis of Mn, Fe, Co, Ni, Cu, Zn, Cd, and Pb in seawater using the Nobias-chelate PA1 resin and magnetic sector inductively coupled plasma mass spectrometry (ICP-MS), Marine Chemistry, 130, 12-20, doi: 10.1016/j.marchem.2011.12.001, 2012.

*5. Section 4.2: It is very difficult to follow the proposed conversion efficiency index (NCEI). It may be a good tool to specifically understand the role of nutrients on nitrogen consumption or productivity, but need to be elaborated more. It is not clear, why summation of differences of treatment and control for consecutive days are used?*
*The proposed conversion efficiency index (NCEI).*
Response:

In the revision, we have clarified the net conversion efficiency index (NCEI) proposed in this study to be an approximate estimation for the utilization of N for the growth of phytoplankton. Therefore, the capacity to synthesize Chl *a* per unit concentration of nitrogen (N) in different treatments can be compared. We agree that the sum consecutive differences over time in Chl *a* concentration between treatments and control will lead to an overestimation of the real net conversion efficiency because of the accumulation effect. Theoretically, the use of the maximum difference will lead to an underestimation of the real net conversion efficiency because of degradation of Chl *a* in the growth period. The real net conversion efficiency should be between those calculated by the two approaches.

We have added a more elaborated description of NCEI in the Methods Section 2.6. (Line 174-181)

*6. Section 4.3: Line 408-419: This paragraph is mostly speculative and difficult to follow, although authors have concluded the importance of DOP determination in seawater.*
Response: The part is particularly important because it provides a new insight to analyse the budget of bioavailable P in the AM-dust addition incubation experiments. Unfortunately, the original presentation seemed to be unclear and didn't service the target well. We have rewritten this paragraph to make our thoughts understandable. (Line 453-466)

Minor comments:

*7. Line 181: should be Table 1.*
Response:
Corrected. (Line 206)

*8. The legends used in Fig. 2, 3 and 4 for nutrients (other than AM-dust) are in same colors and very hard to make out. Most of them are superimposed. It will be useful for reader if different coloured legends with connecting lines can be used.*
Response:
Corrected. (Line 710-720)

[revised manuscript text omitted]
, the aging process of dust followed Guieu's (2010) method and aimed at stimulating the cloud reaction between dust and synthetic evaporating cloud water. The pH around dust in the cloud process (i.e. mix with evaporating cloud water) was found to be as low as ~1 during their transport to the Yellow Sea (Meskhidze et al., 2003), whereas the typical pH in rainwater is 5 (Watanabe et al. 2001, Sasakawa and Uematsu, 2002, Wang et al. 2002, Sakihama et al. 2008, Zhang et al. 2011), meaning that a dilution by a factor of $10e^4$. In consequent, in order to reproduce an evaporating cloud, we have used a concentration that is 10 000-fold larger in our experiments than the typical concentrations found in rainwater. Considering the typical concentrations of dust in rainwaters was 10 mg $L^{-1}$ (Ridame et al., 2002), the dust loading in evaporating cloud water could reach 100 g $L^{-1}$. As a consequence, all of the concentrations in evaporating cloud water were around 10000-fold larger (i.e. 4 orders of magnitude larger) than those in natural rainwater. Table S1 summarized the primary chemical composition of rains in the Eastern Asian regions and the evaporating cloud water used for our simulation. As the uptake of organic acidic gases during transport is complicated for Asian dust, we did not add oxalic acid, which was used for simulating the Saharan dust by Guieu et al. (2010), to simplify the reaction of dust surface and emphasize the importance of inorganic acids ($H_2SO_4$ and $HNO_3$) (Fan et al., 2006; Formenti et al., 2011; Shi et al., 2012).

**References**

[revised manuscript text omitted]

---

## Author Response (AR2)

*Re: "Phytoplankton growth responses to Asian dust additions in the Northwest Pacific Ocean versus the Yellow Sea" (doi: 10.5194/bg-2017-176) by Chao Zhang et al.*

Dear Prof. Manmohan Sarin,

Enclosed is the revised manuscript, "Phytoplankton growth responses to Asian dust additions in the Northwest Pacific Ocean versus the Yellow Sea" (doi: 10.5194/bg-2017-176) by Chao Zhang et al.

In this revision, we tried our best to address all comments and suggestions accordingly. We additionally took this opportunity to correct some technical mistakes through the whole MS.

We thank you again for your consideration of this manuscript.

Sincerely,

Huiwang Gao

Corresponding author

College of Environmental Science and Engineering

Ocean University of China,

Qingdao 266100, China

Phone: +86-532-66782935

Email: hwgao@ouc.edu.cn

**Additional changes:**

 Added an affiliation for the author 'Huiwang Gao'.

 Added ':' at the end of the sentence 'Because of $C_{N:P}$ in AM-dust treatments $\leqslant$ $C_{N:P}$ in N treatments, an inquality can be obtained as below';

 Changed '$CP_{AD}$' to '$CAD_P$', '$CN_{AD}$' to '$CAD_N$', '$CP_N$ to '$CN_P$' to illustrate them clearly;

 Corrected '0.11' $\mu mol\ L^{-1}$ to '0.10 $\mu mol\ L^{-1}$' resulted from the change in the number of decimal places.

**Response to reviews**

**Referee #4**

*The supply of nutrients to the ocean surface layer via dust deposition, and the consequences for phytoplankton growth, have attracted a great deal of attention in recent years. This study focuses on coastal waters that receive dust from the Gobi desert, and compares incubations with added nutrients and incubations with dust that has been treated to mimic the conditions encountered during atmospheric transport. The five stations studied show differing responses, and quantitative interpretation of the results is very challenging.*

Response: We would like to thank the referee very much for the valuable comments which enabled us to improve the quality of the manuscript. We have revised the manuscript accordingly to address the comments.

*The authors have revised the manuscript after review, and have responded satisfactorily to most of the points raised. As Reviewer #2 points out, the observations of co-limitation are not surprising (but do add to the volume of observations on this topic). More interesting is the conclusion that the addition of the modified dust contributed to the mobilisation of otherwise unavailable organic phosphorus. The evidence for this is indirect, and the argument in section 4.3 is not easy to follow. In order to clarify the argument, I suggest that the authors provide a graphical presentation of the numbers presented in lines 440-451. The conclusion of phosphorus mobilisation must be treated as speculative since no measurements of organic phosphorus (DOP and POP) were made; this should emerge as a recommendation for future studies so that this question can be investigated more fully.*

Response: We thank the referee very much for the valuable comment. We have added a graphical presentation in the manuscript, please see newly added Fig. 9. (Line 451, Line 745)

We agree with the reviewer' argument that the conclusion of phosphorus mobilisation must be treated as speculative and emerge as a recommendation for future studies. We revised the manuscript accordingly:

Changed 'The lack of data increased the uncertainty in data interpretation' to 'The lack of data increased data

interpretation to be speculative to some extent'; (Line 467)

Changed 'This suggests that' to 'We speculate that'. (Line 484)

*Reviewers #2 and #3 have in different ways questioned the discussions on trace metals given that only very limited data are available. The discussion of the potential role for Co in phosphorus mobilisation at the end of section 4.3 is unduly speculative at this stage since this process remains to be clearly identified. This discussion should be deleted. In this context, the statement in the abstract that "other micro-constituents" contribute to phytoplankton growth following AM-dust addition is not supported by the experimental results, and should be deleted.*
Response: Correct. (Line 22, Line 462)

*An important question raised by Reviewer #1 is the meaning of "successive increase". This term is introduced in section 2.6, without explaining exactly what is meant. In Figure S1, the dotted lines identify the "successive increase during the incubation period". These are placed on different days at each station, and are not consistently placed at the maximum chlorophyll value in any individual treatment. The definition of this "successive increase" period is clearly of great importance, since the subsequent data analysis focuses on this period. The authors must provide a clear definition of the term "successive increase".*
Response:
We have added a clear definition of the term "successive increase" in Sect. 2.6, Line 168-170, i.e., '
[revised manuscript text omitted]

745

---

## Author Response (AR3)

*Re: "Phytoplankton growth response to Asian dust addition in the Northwest Pacific Ocean versus the Yellow Sea" (doi: 10.5194/bg-2017-176) by Chao Zhang et al.*

Dear Prof. Manmohan Sarin,

Enclosed is the revised manuscript, "Phytoplankton growth response to Asian dust addition in the Northwest Pacific Ocean versus the Yellow Sea" (doi: 10.5194/bg-2017-176) by Chao Zhang et al.

In this revision, we have considered and followed the suggested changes.

We thank you again for your consideration of this manuscript.

Sincerely,

Huiwang Gao

Corresponding author

College of Environmental Science and Engineering

Ocean University of China,

Qingdao 266100, China

Phone: +86-532-66782935

Email: hwgao@ouc.edu.c

**Response to reviews**

*Authors may consider following changes while submitting a revised version of the manuscript.*
Response: We would like to thank the Associate Editor very much for the valuable and considerate comments which enabled us to improve the expression of the manuscript. We have revised the manuscript accordingly to address the comments.

*Lines 167-168: We focused on analyzing the initial 2–5 days when Chl a concentration shows steady increase during the incubation period.*
Response: Correct. (Line 167-168)

*Lines 169-170: Thus, we define the initial five days as successive increase in Chl a concentration at S1. The same definition is also applicable for other stations.*
Response: Correct. (Line 169-171)

*Lines 465-468: However, no measurements of trace metals in seawater as well as DOP and P in the phytoplankton cells were made in our study. The lack of data on measurements of organic P makes the interpretation rather speculative but serves as a recommendation for future studies.*
Response: Correct. (Line 465-468)

*Line 484: We suggest that there are other sources of P during the incubation*
Response: Correct. (Line 484)

*Title: Phytoplankton growth response to Asian dust addition in the Northwest Pacific Ocean versus the Yellow Sea*
Response: Correct. (Line 1-2)

[revised manuscript text omitted]